# ReFT: Representation Finetuning
# for Language Models

**Zhengxuan Wu**[*][†]    **Aryaman Arora**[*][†]    **Zheng Wang**[†]    **Atticus Geiger**[‡]
**Dan Jurafsky**[†]    **Christopher D. Manning**[†]    **Christopher Potts**[†]
[†]Stanford University    [‡]Pr(Ai)$^2$R Group
{wuzhengx,aryamana,peterwz,atticusg}@stanford.edu
{jurafsky,manning,cgpotts}@stanford.edu

## Abstract

Parameter-efficient finetuning (PEFT) methods seek to adapt large neural models via updates to a small number of *weights*. However, much prior interpretability work has shown that *representations* encode rich semantic information, suggesting that editing representations might be a more powerful alternative. We pursue this hypothesis by developing a family of **Representation Finetuning (ReFT)** methods. ReFT methods operate on a frozen base model and learn task-specific interventions on hidden representations. We define a strong instance of the ReFT family, Low-rank Linear Subspace ReFT (LoReFT), and we identify an ablation of this method that trades some performance for increased efficiency. Both are drop-in replacements for existing PEFTs and learn interventions that are 15×–65× more parameter-efficient than LoRA. We showcase LoReFT on eight commonsense reasoning tasks, four arithmetic reasoning tasks, instruction-tuning, and GLUE. In all these evaluations, our ReFTs deliver the best balance of efficiency and performance, and almost always outperform state-of-the-art PEFTs. We release a generic ReFT training library publicly at `https://github.com/stanfordnlp/pyreft`.

## 1 Introduction

Pretrained language models (LMs) are frequently finetuned to adapt them to new domains or tasks [Dai and Le, 2015]. With finetuning, a single base model can be adapted to a variety of tasks given only small amounts of in-domain data. However, finetuning large LMs is expensive. Parameter-efficient finetuning (PEFT) methods propose to address the high costs of full finetuning by updating a small number of weights. This reduces memory usage and training time, and PEFTs achieve similar performance to full finetuning in many settings [Hu et al., 2023].

A hallmark of current state-of-the-art PEFTs is that they modify *weights* rather than *representations*. However, much prior interpretability work has shown that representations encode rich semantic information, suggesting that editing representations might be a more powerful alternative to weight updates. In this paper, we pursue this hypothesis by developing and motivating **Representation Finetuning (ReFT)**. Instead of adapting model weights, ReFT methods train interventions that manipulate a small fraction of model representations in order to steer model behaviours to solve downstream tasks at inference time. ReFT methods are drop-in replacements for weight-based PEFTs. This approach is inspired by recent work in LM interpretability that intervenes on representations to find faithful causal mechanisms [Geiger et al., 2023] and to steer model behaviours at inference time [Turner et al., 2023, Li et al., 2024], and it can be seen as a generalisation of the representation-editing work of Wu et al. [2024a], Turner et al. [2023], and Zou et al. [2023] (see appendix B for details).

We focus on a strong and highly efficient instance of the ReFT family that we call **Low-rank Linear Subspace ReFT** (**LoReFT**). LoReFT is a parametrisation of ReFT that intervenes on hidden

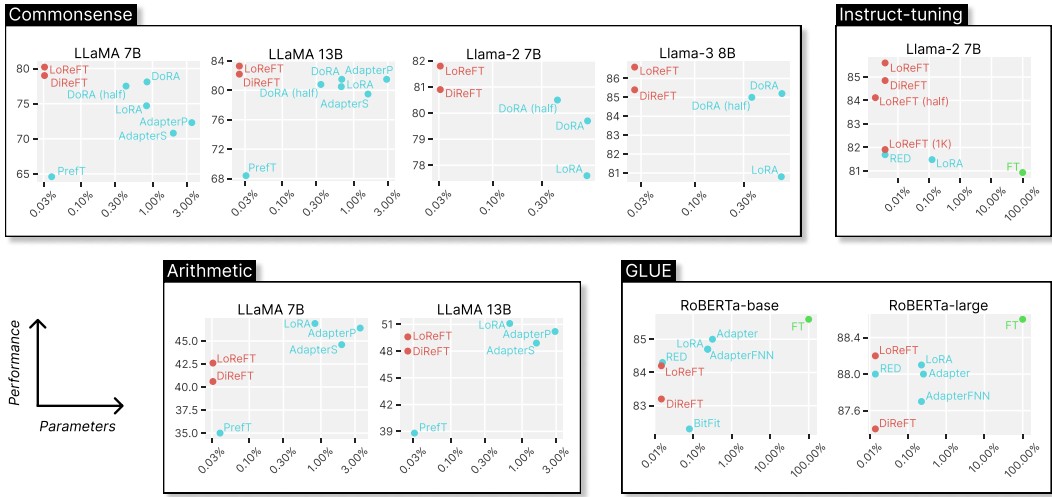

Figure 1: Parameter count vs. performance for LoReFT and other PEFTs across four benchmarks when applied to LLaMA, Llama-2, Llama-3, and RoBERTa models. Despite training far fewer parameters than existing PEFTs, LoReFT achieves competitive or even state-of-the-art performance on all tasks. Its value is most apparent for the largest models in our evaluations. **Note**: FT is full-parameter finetuning, which is not a PEFT or ReFT method. Additional results are in section 4.

representations in the linear subspace spanned by a low-rank projection matrix, building directly on the distributed alignment search (DAS) method of Geiger et al. [2023] and Wu et al. [2023]. We also identify an ablation of this method (DiReFT) that trades some performance for increased efficiency. We evaluate our ReFTs on LLaMA-family models and small-scale LMs against existing PEFTs on standard benchmarks from four domains: commonsense reasoning, arithmetic reasoning, instruction-following, and natural language understanding. Compared to LoRA, we find that LoReFT uses 15×–65× times fewer parameters while achieving state-of-the-art performance on commonsense reasoning, instruction-following, and natural language understanding against the strongest PEFTs. These findings indicate that ReFT methods are worthy of further exploration, as they may emerge as more efficient and effective alternatives to weight-based PEFTs.

## 2 Related work

**Parameter-efficient finetuning methods (PEFTs).** PEFTs train a fraction of the model's parameters to adapt it to downstream tasks. We classify PEFTs into three categories:

1. **Adapter-based methods** train additional modules (e.g. fully-connected layers) on top of the frozen pretrained model. *Series adapters* insert components between LM attention or MLP layers [Houlsby et al., 2019, Pfeiffer et al., 2020, Wang et al., 2022, He et al., 2022b, Fu et al., 2021], while *parallel adapters* add modules alongside existing components [He et al., 2022a]. Since adapters add new components that cannot be easily folded into existing model weights, they impose an additional burden at inference time.[1]

2. **LoRA** [Hu et al., 2022] and DoRA [Liu et al., 2024c] use low-rank matrices to approximate additive weight updates during training, and require no additional overhead during inference since the weight updates can be merged into the model. These are the strongest PEFTs currently.[2]

3. **Prompt-based methods** add randomly-initialised soft tokens to the input (usually as a prefix) and train their embeddings while keeping the LM weights frozen [Li and Liang, 2021]. These methods are often far from optimal compared to other PEFTs, and come at the cost of significant

---

[1]Several very recent papers introduce new adapter architectures but do not benchmark them on the tasks we consider, or they perform hyperparameter-tuning in a different setup than done in this work. These include: LLaMA-Adapter [Zhang et al., 2024b], LLaMA-Adapter v2 [Gao et al., 2023], Aligner [Ziheng et al., 2023].

[2]Additional methods not studied in this work: AutoLoRA [Zhang et al., 2024c], ResLoRA [Shi et al., 2024], SiRA [Zhu et al., 2023].

inference overhead. A variant of this method where hidden-layer activations are also tuned was introduced as a baseline in Hu et al. [2022], with better performance.

**Representation editing.** Recent work on *activation steering* and *representation engineering* shows that adding fixed or task-specific steering vectors [Subramani et al., 2022, Turner et al., 2023, Zou et al., 2023, Liu et al., 2024b, Vogel, 2024, Li et al., 2024] or applying concept erasure [Ravfogel et al., 2022, Belrose et al., 2023, Avitan et al., 2024, Singh et al., 2024] to the residual stream can enable a degree of control over pretrained LM generations without the need for resource-intensive finetuning [Wu et al., 2024a]. The success of these methods affirms that representations induced by pretrained LMs carry rich semantic structure.

**Interventional interpretability.** Much recent work has used interventions on model-internal states to test hypotheses about how LMs implement various behaviours. In particular, interventions on linear subspaces of representations have provided increasing evidence that human-interpretable concepts are encoded linearly [Smolensky, 1986, Rumelhart et al., 1986, McClelland et al., 1986]. This includes linguistic features such as gender and number [Lasri et al., 2022, Wang et al., 2023, Hanna et al., 2023, Chintam et al., 2023, Yamakoshi et al., 2023, Hao and Linzen, 2023, Chen et al., 2023, Amini et al., 2023, Guerner et al., 2023, Arora et al., 2024], logical and mathematical reasoning [Wu et al., 2023], entity attributes [Huang et al., 2024], and a number of other domains [Mikolov et al., 2013, Elhage et al., 2022, Park et al., 2023, Nanda et al., 2023, Guerner et al., 2023].

# 3   ReFT

We now define the ReFT family of methods. To do this, we first summarize the core motivation, which emerges from work on intervention-based model interpretability. We then show how this leads directly to Low-rank Linear Subspace ReFT (LoReFT). Finally, we generalize this to a family of ReFT methods. Appendix A provides a brief overview of our generic ReFT training library.

To keep the presentation simple, we assume throughout that our target model is a Transformer-based [Vaswani et al., 2017] LM that produces contextualised representations of sequences of tokens. Given a sequence of $n$ input tokens $\mathbf{x} = (x_1, \dots, x_n)$, the model first embeds these into a list of representations $\mathbf{h}^{(0)} = (\mathbf{h}_1^{(0)}, \dots, \mathbf{h}_n^{(0)})$. Then, $m$ layers successively compute the $j$-th list of hidden representations $\mathbf{h}^{(j)}$ as a function of the previous list of hidden representations $\mathbf{h}^{(j-1)}$. Each hidden representation is a vector $\mathbf{h} \in \mathbb{R}^d$. The LM uses the final hidden representations $\mathbf{h}^{(m)}$ to produce its predictions. In our experiments, we consider both autoregressive LMs and masked LMs [Devlin et al., 2019]. An autoregressive LM predicts $p(x_{n+1} \mid x_1, \dots, x_n) = \mathrm{softmax}(\mathbf{W}\mathbf{h}_n^{(m)})$, while a masked LM predicts $p(x_i \mid x_1, \dots, x_{i-1}, x_{i+1}, \dots, x_n) = \mathrm{softmax}(\mathbf{W}\mathbf{h}_i^{(m)})$, where $\mathbf{W}$ is a learned matrix mapping from representations to logits over the vocabulary space.

## 3.1   Motivation

In interpretability research, the framework of causal abstraction [Geiger et al., 2021] uses **interchange interventions** to establish the causal role of representations in deep learning models. An interchange intervention fixes a representation to the value it would take if a counterfactual input were processed by the model. Experiments investigating how such interventions affect model behavior form the evidence for claims about the causal role of a representation and the concept it encodes.

To test whether a concept is encoded in a linear subspace of a representation, one may use a **distributed interchange intervention** (DII) [Geiger et al., 2023].[3] Let $\mathbf{h}_b$ be the hidden representation created at row $i$ and column $k$ when our model processes input $b$, and let $\mathbf{h}_s$ be the corresponding representation when that same model processes input $s$. A distributed interchange intervention on $\mathbf{h}_b$ given a counterfactual source representation $\mathbf{h}_s$ is then defined as

$$\mathrm{DII}(\mathbf{h}_b, \mathbf{h}_s, \mathbf{R}) = \mathbf{b} + \mathbf{R}^\top(\mathbf{R}\mathbf{h}_s - \mathbf{R}\mathbf{h}_b) \tag{1}$$

where $\mathbf{R} \in \mathbb{R}^{r \times d}$ is a low-rank projection matrix with orthonormal rows, $d$ is the representation dimensionality, and $r$ is the dimensionality of the subspace we are intervening on. We learn the subspace $\mathbf{R}$ using distributed alignment search (DAS), which finds the subspace that maximises the probability of the expected counterfactual output after intervention [Geiger et al., 2023]. DAS is

---

[3]This notion of subspace intervention was also independently discovered by Guerner et al. [2023].

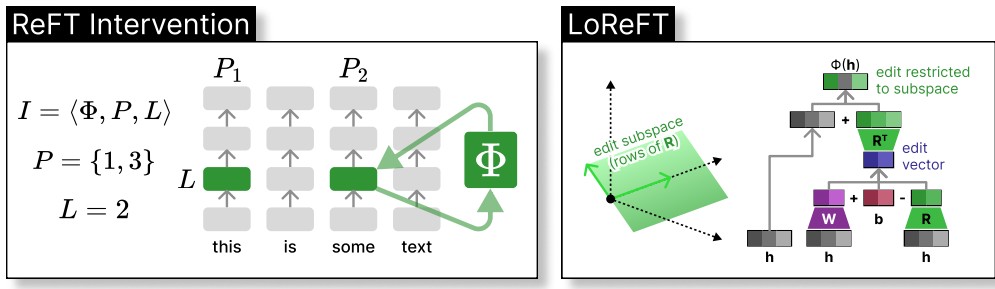

Figure 2: **Illustration of ReFT**. **(1)** The left panel depicts an intervention $I$: the intervention function $\Phi$ is applied to hidden representations at positions $P$ in layer $l$. **(2)** The right panel depicts the intervention function used in LoReFT, which finds an edit vector that only modifies the representation in the linear subspace spanned by the rows of $\mathbf{R}$. Specifically, we show how a rank-2 LoReFT operates on 3-dimensional hidden representations.

highly expressive, and can effectively localize concepts within model representations [Wu et al., 2023, Arora et al., 2024, Wu et al., 2024c, Huang et al., 2024]. This suggests that subspace representation interventions could also be a powerful tool for model control.

### 3.2 Two low-rank ReFT instantiations

**LoReFT.** The formulation of DII in eq. (1) immediately suggests a way to control model generations via interventions. The guiding intuition is that we can learn how to perform interventions that steer the model towards predicting our task labels. The resulting method, Low-rank Linear Subspace ReFT (LoReFT), is defined by the following variant of eq. (1):

$$\Phi_{\mathsf{LoReFT}}(\mathbf{h}) = \mathbf{h} + \mathbf{R}^\top \left(\mathbf{W}\mathbf{h} + \mathbf{b} - \mathbf{R}\mathbf{h}\right) \tag{2}$$

This is identical to eq. (1), except we use a *learned projected source* $\mathbf{R}\mathbf{s} = \mathbf{W}\mathbf{h} + \mathbf{b}$. LoReFT thus edits the representation in the $r$-dimensional subspace spanned by the rows of $\mathbf{R}$ to take on the values obtained from our linear projection $\mathbf{W}\mathbf{h} + \mathbf{b}$. We depict this operation in fig. 2. The learned parameters are $\phi = \{\mathbf{R}, \mathbf{W}, \mathbf{b}\}$; the parameters of the LM are frozen. As with DII, $\mathbf{R} \in \mathbb{R}^{r \times d}$ is a low-rank matrix with orthonormal rows where $d$ is the hidden-state dimensionality and $r \leq d$ is the rank of the subspace. We further define a linear projection $\mathbf{W} \in \mathbb{R}^{r \times d}$ and bias vector $\mathbf{b} \in \mathbb{R}^r$.

**DiReFT.** In addition, we define an ablation of LoReFT which removes the orthogonality constraint and the difference operation, reducing training time:

$$\Phi_{\mathsf{DiReFT}}(\mathbf{h}) = \mathbf{h} + \mathbf{W}_2^\top \left(\mathbf{W}_1\mathbf{h} + \mathbf{b}\right) \tag{3}$$

Both $\mathbf{W}_1, \mathbf{W}_2 \in \mathbb{R}^{r \times d}$ are low-rank projection matrices. Note that eq. (3) resembles LoRA, and thus DiReFT can be thought of as LoRA applied **di**rectly to hidden representations at certain positions.[4] Empirical evidence from previous work suggests that adding orthogonal constraints to LoRA weights increases performance [Liu et al., 2024d]. (Appendix E reports results for additional ablations of LoReFT.)

**Training objective.** We consider both generation tasks using decoder-only or encoder–decoder LMs and classification tasks using encoder-only models with $m$ layers. The pretrained language model induces a distribution over token sequences $p(\cdot)$. We denote the model that results from the ReFT intervention $\Phi$ on $p(\cdot)$ as $p_\Phi(\cdot)$ with trainable parameters $\phi$. To simplify notation, we refer to the hidden representations produced by the LM on input $\mathbf{x}$ as $\mathbf{h}(\mathbf{x})$, and those by the intervened LM as $\mathbf{h}_\Phi(\mathbf{x})$.

For generation tasks, our training objective is language modelling. Given an input sequence $\mathbf{x} = (x_1, \dots, x_n)$ with $n$ tokens as the prompt, the goal is to predict the output sequence $\mathbf{y} = (y_1, \dots, y_k)$

---

[4]LoRA is not applicable to the residual stream, which is weightless. LoRA can be configured to apply only to the attention layer output projection matrix, which is similar to our residual stream intervention. However, previous works found that applying LoRA only to attention layers is sub-optimal [Hu et al., 2023].

with $k$ tokens. We minimise the cross-entropy loss with teacher-forcing over all output positions.

$$\min_{\phi} \left\{ -\sum_{i=1}^{k} \log p_\Phi \left( y_i \mid \mathbf{x}\mathbf{y}_{<i} \right) \right\} \tag{4}$$

For single-label classification tasks, we add a classification head $H_\theta(\cdot)$ with parameters $\theta$ that takes the final-layer (i.e., layer $m$) representation at the first token (CLS) $\mathbf{h}_1^{(m)}$ as input and outputs a distribution over classes. $H$ has the learned parameters $\theta = \{\mathbf{W}_o, \mathbf{b}_o, \mathbf{W}_d, \mathbf{b}_d\}$.

$$H_\theta(\cdot \mid \mathbf{h}) = \mathrm{softmax}\left( \mathbf{W}_o(\tanh(\mathbf{W}_d\mathbf{h}_1^{(m)} + \mathbf{b}_d)) + \mathbf{b}_o \right) \tag{5}$$

We learn the parameters of the head and those of the intervention function $\Phi$. We minimise the cross-entropy loss of the target class $y$ given input $\mathbf{x}$:

$$\min_{\phi,\theta} \{-\log H_\theta(y \mid \mathbf{h}_\Phi(\mathbf{x}))\} \tag{6}$$

### 3.3 The ReFT family of methods

It is straightforward to generalise the above intervention functions to define a family of intervention-based representation finetuning methods. We first define a general notion of *intervention*, i.e. the modification of hidden representations during the model forward pass:

**Definition 3.1.** An **intervention** $I$ is a tuple $\langle \Phi, P, l \rangle$ that encapsulates a single inference-time modification of the representations computed by a Transformer-based LM. The three components of an intervention are (1) the **intervention function** $\Phi : \mathbb{R}^d \to \mathbb{R}^d$ with learned parameters $\phi$, (2) a set of **input positions** $P \subseteq \{1, \ldots, n\}$ that the intervention is applied to, and (3) the **layer** $l \in \{1, \ldots, m\}$ at which the intervention is applied.

We implement the intervention $I$ as the following operation that overwrites some representations $\mathbf{h}$:

$$\mathbf{h}^{(l)} \leftarrow \left( \Phi\left(\mathbf{h}_p^{(l)}\right) \text{ if } p \in P \text{ else } \mathbf{h}_p^{(l)} \right)_{p \in 1,\ldots,n} \tag{7}$$

The intervention is applied immediately after the computation of $\mathbf{h}^{(l)}$ and thus affects the representations computed in later layers $\mathbf{h}^{(l+1)}, \ldots, \mathbf{h}^{(m)}$.

Figure 2 provides a schematic overview of an intervention. A ReFT is then defined as a constrained set of non-overlapping interventions:

**Definition 3.2.** A **ReFT method** is a set of $f$ interventions $\mathbf{I} = \{I_1, \ldots, I_f\}$. We enforce that for any two interventions $I_j, I_k \in \mathbf{I}$ such that they operate on the same layer $l_j = l_k$, their intervention positions must be disjoint, i.e. $P_j \cap P_k = \varnothing$. The parameters $(\phi_1, \ldots, \phi_f)$ of all of the intervention functions are independent.

ReFT is thus a generic framework encompassing interventions on hidden representations during the model forward pass. In appendix B, we show how a variety of existing inference-time intervention methods can be described within this framework.

## 4 Experiments

To evaluate our ReFTs against existing PEFTs, we conduct experiments across four diverse NLP benchmarks covering more than 20 datasets (extensive details on our datasets are in appendix C). Our goal is to provide a rich picture of how LoReFT and DiReFT perform in different scenarios. We experiment with both masked and autoregressive LMs at different scales, ranging from RoBERTa-base [Liu et al., 2019] with 125M to LLaMA models [Touvron et al., 2023a,b] with 13B parameters. We benchmark against existing PEFTs such as prefix-tuning [Li and Liang, 2021], adapter-tuning with both Series Adapters and Parallel Adapters, BitFit [Ben Zaken et al., 2022], RED [Wu et al., 2024a], LoRA [Hu et al., 2022], and DoRA [Liu et al., 2024c]. Our comparisons focus on both performance and parameter efficiency. In our comparisons, we use hyperparameter-tuned scores from previous works when possible. We load our base LMs in torch.bfloat16 to save memory. **All of our experiments are run with a single GPU: NVIDIA A100 40G/80G or RTX 6000**. Examples of raw model generations are in appendix I. The performance results of all baseline methods are adapted from Liu et al. [2024c] and represent the best performance achieved after hyperparameter tuning.

## 4.1 Hyperparameter configuration

For our experiments, we must decide how many interventions to learn and which layers and input positions to apply each one on. We propose learning interventions on a fixed number of $p$ prefix and $s$ suffix positions in the prompt. Specifically, we tune four hyperparameters:

1. The number of prefix positions $p$ to intervene on, i.e. positions $\{1, \ldots, p\}$.
2. The number of suffix positions $s$ to intervene on, i.e. positions $\{n - s + 1, \ldots, n\}$.
3. Which set of layers $L$ to intervene on.
4. Whether or not to tie intervention parameters $\phi$ across different positions in the same layer.

This simplifies the hyperparameter search space; compared to LoRA, the only additional consideration is which positions to intervene on. Since the number of positions edited is constant, LoReFT and DiReFT contribute a fixed additional inference cost that does not scale with prompt length.

Given the positions $P = \{1, \ldots, p\} \cup \{n - s + 1, \ldots, n\}$, we define the untied and tied variants:

$$\mathbf{I}_{\text{untied}} = \{\langle \Phi, \{p\}, l \rangle \mid p \in P, l \in L\} \qquad \mathbf{I}_{\text{tied}} = \{\langle \Phi, P, l \rangle \mid l \in L\}$$

Additionally, when applying LoReFT and DiReFT to a prompt with length $n$ where $n < p + s$, we set $p \leftarrow \min(p, \lfloor n/2 \rfloor)$ and $s \leftarrow \min(s, \lceil n/2 \rceil)$ and do not apply the truncated interventions in $\mathbf{I}_{\text{untied}}$. We also tune neural-network training hyperparameters.

Unlike previous work [Hu et al., 2022, 2023, Liu et al., 2024c] where hyperparameter tuning may involve optimising performance directly on test sets, we only tune our hyperparameters on development sets which do not contain any overlapping examples with the test sets of our tasks. We further describe hyperparameter tuning for each benchmark in appendix D.1.

## 4.2 Commonsense reasoning

We replicate the experimental setup in Hu et al. [2023] and finetune LLaMA-1 7B/13B, Llama-2 7B, and Llama-3 8B[5] on a combined dataset of eight commonsense reasoning tasks (COMMONSENSE170K). We report scores on each task's test set individually. We compare with PEFTs benchmarked in Hu et al. [2023] as well as the identical experiment reported in Liu et al. [2024c] for DoRA.

**Datasets.** Our benchmark contains eight commonsense reasoning datasets, including BoolQ [Clark et al., 2019], PIQA [Bisk et al., 2020], SIQA [Sap et al., 2019], HellaSwag [Zellers et al., 2019], WinoGrande [Sakaguchi et al., 2021], ARC-e, ARC-c [Clark et al., 2018], and OBQA [Mihaylov et al., 2018]. Examples are formulated as multiple-choice problems where the model needs to directly generate the correct choice without rationales. We use the same prompt template as in Hu et al. [2023] with additional string normalisation (removing leading and trailing whitespace).

**Hyperparameter tuning.** We do not do hyperparameter selection based on test set results. Rather, we use the hyperparameter settings of the model that performs best on a development set created from the GSM8K training set, except we use a lower number of epochs (6 instead of 12) because the COMMONSENSE170K training set is more than 20 times larger than GSM8K. This allows us to tune relevant hyperparamters, and also serves to test the robustness of these settings across different domains. We additionally report scores on 3 epochs in appendix D.3.

**Results.** We report results in table 1. LoReFT sets state-of-the-art performance on the commonsense reasoning tasks, outperforming all other methods by a considerable margin. While being more compute-efficient, DiReFT achieves only slightly worse performance consistently.

## 4.3 Arithmetic reasoning

Similar to the previous experiment, we follow the experimental setup in Hu et al. [2023] and finetune LLaMA-1 7B and 13B on a combined dataset of seven arithmetic reasoning tasks with LM-generated chain-of-thought steps (MATH10K) and report scores on four of the tasks' test sets. We only evaluate correctness on the final numeric or multiple-choice answer.

---

[5]Llama-3 8B appeared on April 18, 2024, and thus we had time to complete only commonsense reasoning experiments with this model. Liu et al. [2024c] report corresponding results for LoRA and DoRA.

Table 1: Accuracy comparison of LLaMA-1 7B/13B, Llama-2 7B and Llama-3 8B against existing PEFT methods on eight commonsense reasoning datasets. *Performance results of all baseline methods are taken from Liu et al. [2024c]. We report averaged performance of three runs with distinct random seeds for our method. For our methods, Param. (%) is calculated by dividing the number of trainable parameters by the number of parameters of the base LM.

| Model | PEFT | Params (%) | Accuracy (↑) | | | | | | | | |
|---|---|---|---|---|---|---|---|---|---|---|---|
| | | | BoolQ | PIQA | SIQA | HellaS. | WinoG. | ARC-e | ARC-c | OBQA | Avg. |
| ChatGPT* | — | — | 73.1 | 85.4 | 68.5 | 78.5 | 66.1 | 89.8 | 79.9 | 74.8 | 77.0 |
| LLaMA-7B | PrefT* | 0.039% | 64.3 | 76.8 | 73.9 | 42.1 | 72.1 | 72.9 | 54.0 | 60.6 | 64.6 |
| | Adapter$^S$* | 1.953% | 63.0 | 79.2 | 76.3 | 67.9 | 75.7 | 74.5 | 57.1 | 72.4 | 70.8 |
| | Adapter$^P$* | 3.542% | 67.9 | 76.4 | 78.8 | 69.8 | 78.9 | 73.7 | 57.3 | 75.2 | 72.3 |
| | LoRA* | 0.826% | 68.9 | 80.7 | 77.4 | 78.1 | 78.8 | 77.8 | 61.3 | 74.8 | 74.7 |
| | DoRA (half)* | 0.427% | **70.0** | 82.6 | 79.7 | 83.2 | 80.6 | 80.6 | 65.4 | 77.6 | 77.5 |
| | DoRA* | 0.838% | 68.5 | 82.9 | 79.6 | 84.8 | 80.8 | 81.4 | 65.8 | **81.0** | 78.1 |
| | **DiReFT (ours)** | 0.031% | 69.5 | 83.0 | 79.0 | 92.5 | 80.5 | 82.2 | 68.0 | 77.5 | 79.0 |
| | **LoReFT (ours)** | 0.031% | 69.3 | **84.4** | **80.3** | **93.1** | **84.2** | **83.2** | **68.2** | 78.9 | **80.2** |
| LLaMA-13B | PrefT* | 0.031% | 65.3 | 75.4 | 72.1 | 55.2 | 68.6 | 79.5 | 62.9 | 68.0 | 68.4 |
| | Adapter$^S$* | 1.586% | 71.8 | 83.0 | 79.2 | 88.1 | 82.4 | 82.5 | 67.3 | 81.8 | 79.5 |
| | Adapter$^P$* | 2.894% | **72.5** | 84.9 | 79.8 | 92.1 | 84.7 | 84.2 | 71.2 | 82.4 | 81.5 |
| | LoRA* | 0.670% | 72.1 | 83.5 | 80.5 | 90.5 | 83.7 | 82.8 | 68.3 | 82.4 | 80.5 |
| | DoRA (half)* | 0.347% | **72.5** | 85.3 | 79.9 | 90.1 | 82.9 | 82.7 | 69.7 | 83.6 | 80.8 |
| | DoRA* | 0.681% | 72.4 | 84.9 | 81.5 | 92.4 | 84.2 | 84.2 | 69.6 | 82.8 | 81.5 |
| | **DiReFT (ours)** | 0.025% | 71.3 | 86.1 | 80.8 | 94.6 | 83.6 | 85.5 | 72.9 | 82.7 | 82.2 |
| | **LoReFT (ours)** | 0.025% | 72.1 | **86.3** | **81.8** | **95.1** | **87.2** | **86.2** | **73.7** | **84.2** | **83.3** |
| Llama-2 7B | LoRA* | 0.826% | 69.8 | 79.9 | 79.5 | 83.6 | 82.6 | 79.8 | 64.7 | 81.0 | 77.6 |
| | DoRA (half)* | 0.427% | **72.0** | 83.1 | 79.9 | 89.1 | 83.0 | 84.5 | 71.0 | 81.2 | 80.5 |
| | DoRA* | 0.838% | 71.8 | 83.7 | 76.0 | 89.1 | 82.6 | 83.7 | 68.2 | **82.4** | 79.7 |
| | **DiReFT (ours)** | 0.031% | 70.8 | 83.6 | 80.2 | 93.6 | 82.1 | 84.8 | 70.4 | 81.5 | 80.9 |
| | **LoReFT (ours)** | 0.031% | 71.1 | **83.8** | **80.8** | **94.3** | **84.5** | **85.6** | **72.2** | 82.3 | **81.8** |
| Llama-3 8B | LoRA* | 0.700% | 70.8 | 85.2 | 79.9 | 91.7 | 84.3 | 84.2 | 71.2 | 79.0 | 80.8 |
| | DoRA (half)* | 0.361% | 74.5 | 88.8 | 80.3 | 95.5 | 84.7 | 90.1 | 79.1 | 87.2 | 85.0 |
| | DoRA* | 0.710% | 74.6 | 89.3 | 79.9 | 95.5 | 85.6 | 90.5 | 80.4 | 85.8 | 85.2 |
| | **DiReFT (ours)** | 0.026% | 73.4 | 88.7 | 81.0 | 95.6 | 85.5 | 91.8 | **81.8** | 85.4 | 85.4 |
| | **LoReFT (ours)** | 0.026% | **75.1** | **90.2** | **82.0** | **96.3** | **87.4** | **92.4** | 81.6 | **87.5** | **86.6** |

**Hyperparameter tuning.** We use the same hyperparameter settings as for the Commonsense Reasoning benchmark, but with 12 epochs for training. We also report scores on 3 epochs.

**Datasets.** Our benchmark contains four datasets for math world problems, including AQuA [Ling et al., 2017], GSM8K [Cobbe et al., 2021], MAWPS [Koncel-Kedziorski et al., 2016], and SVAMP [Patel et al., 2021]. Models need to generate chain-of-thought [Wei et al., 2022] before the final answer. We use the same prompt template and hyperparameter settings as in the previous experiment.

**Results.** We report results in table 2. We find that both LoReFT and DiReFT do not perform as well at arithmetic reasoning tasks compared to LoRA and adapters, but do outperform prefix-tuning. Our results suggest that our ReFTs may have more trouble on chain-of-thought reasoning than the single-step commonsense reasoning tasks due to the length of generations (greater length necessarily reduces the effect of the intervention) and overall greater difficulty of the task. Our results show that our ReFTs perform better with the 13B model than the 7B model, which suggests that our methods scale with model size. Overall, we note that the arithmetic reasoning results show a lot of variation, with no single method emerging as a clear winner across all of them.

## 4.4 Instruction-following

Base LMs require instruction finetuning to follow human prompts [Ouyang et al., 2022]. We follow the experimental setup in Wu et al. [2024a] and finetune Llama-2 7B with Ultrafeedback [Cui et al., 2023]. We compare against full parameter finetuning, LoRA, and RED. For evaluation, we use Alpaca-Eval v1.0 [Li et al., 2023], which computes the win-rate against text-davinci-003 using GPT-4 as the annotator. We use the same prompt template as in Taori et al. [2023].

**Datasets.** Ultrafeedback is high-quality instruction dataset where responses are generated via scoring a diverse set of model responses from a list of candidates (e.g. ChatGPT and Bard). The score is calculated as a weighted score of instruction-following, truthfulness, honesty, and helpfulness.

Table 2: Accuracy comparison of LLaMA-1 7B/13B against existing PEFT methods on four arithmetic reasoning datasets. *Performance results of all baseline methods are taken from Hu et al. [2023]. We report averaged performance of three runs with distinct random seeds for our method.

| Model | PEFT | Params (%) | Accuracy (↑) | | | | |
|-------|------|------------|------|-------|-------|-------|------|
| | | | AQuA | GSM8K | MAWPS | SVAMP | Avg. |
| LLaMA-7B | PrefT* | 0.039% | 14.2 | 24.4 | 63.4 | 38.1 | 35.0 |
| | Adapter^S* | 1.953% | 15.0 | 33.3 | 77.7 | **52.3** | 44.6 |
| | Adapter^P* | 3.542% | 18.1 | 35.3 | **82.4** | 49.6 | 46.4 |
| | LoRA* | 0.826% | 18.9 | **37.5** | 79.0 | 52.1 | **46.9** |
| | **DiReFT (ours)** | 0.031% | 21.3 | 24.1 | 74.5 | 42.7 | 40.6 |
| | **LoReFT (ours)** | 0.031% | **21.4** | 26.0 | 76.2 | 46.8 | 42.6 |
| LLaMA-13B | PrefT* | 0.031% | 15.7 | 31.1 | 66.8 | 41.4 | 38.8 |
| | Adapter^S* | 1.586% | 22.0 | 44.0 | 78.6 | 50.8 | 48.9 |
| | Adapter^P* | 2.894% | 20.5 | 43.3 | 81.1 | **55.7** | 50.2 |
| | LoRA* | 0.670% | 18.5 | **47.5** | **83.6** | 54.6 | **51.1** |
| | **DiReFT (ours)** | 0.025% | 20.5 | 35.8 | 80.8 | 54.8 | 48.0 |
| | **LoReFT (ours)** | 0.025% | **23.6** | 38.1 | 82.4 | 54.2 | 49.6 |

Table 3: Instruction tuning evaluation results for instruction-tuned Llama-2 7B with Alpaca-Eval v1.0. We report averaged performance of two runs with distinct random seeds for our method. *half* denotes our runs with half of the rank; *1K* denotes our runs with a low-resource setting where there is only 1K training examples. [†]Performance results of baseline methods are taken from Li et al. [2023]. *Performance results of baseline methods are taken from Wu et al. [2024a]. [‡]**It takes 18 minutes to train our Llama-2 Chat 7B on 1K examples using a single A100 40G GPU with ≈1MB parameters on disk.**

| Model & PEFT | Params (%) | Win-rate (↑) |
|--------------|------------|--------------|
| GPT-3.5 Turbo 1106[†] | — | 86.30 |
| Llama-2 Chat 13B[†] | — | 81.10 |
| Llama-2 Chat 7B[†] | — | 71.40 |
| Llama-2 7B & FT* | 100% | 80.93 |
| Llama-2 7B & LoRA* | 0.1245% | 81.48 |
| Llama-2 7B & RED* | 0.0039% | 81.69 |
| Llama-2 7B & **DiReFT (ours)** | 0.0039% | 84.85 |
| Llama-2 7B & **LoReFT (ours)** | 0.0039% | **85.60** |
| Llama-2 7B & **LoReFT (ours, *half*)** | 0.0019% | 84.12 |
| Llama-2 7B & **LoReFT (ours, *1K*)**[‡] | 0.0039% | 81.91 |

Some of the best 7B and 13B chat-models (e.g. UltraLM-13B [Ding et al., 2023]) are finetuned with Ultrafeedback.

**Hyperparameter tuning.** We do hyperparameter-tuning on the unseen instruction-following dataset Alpaca-52K [Taori et al., 2023] with only LLaMA-7B to prevent test-set hill-climbing. We then use the hyperparameter settings of our best performing model to finetune on Ultrafeedback. For hyperparameter tuning, we use Alpaca-Eval v1.0 with GPT-4 turbo as the annotator for fast turnaround, which also prevents overfitting with GPT-4 as a judge.

**Results.** We report results in table 3. When matched in parameter count to the previous most parameter-efficient PEFT (RED) and trained on Llama-2 7B, LoReFT outperforms all reported finetuning methods (including full finetuning) and achieves a win-rate within 1% of GPT-3.5 Turbo 1106. Furthermore, after halving the parameter count or using only 1/64-th of the data, LoReFT still outperforms other finetuning methods. This result shows that LoReFT can succeed at long-form text generation. DiReFT is again slightly worse than LoReFT but is highly competitive.

Table 4: Accuracy comparison of RoBERTa-base and RoBERTa-large against existing PEFT methods on the GLUE benchmark. *Performance results of all baseline methods are taken from Wu et al. [2024a]. We report averaged performance of five runs with distinct random seeds for our method.

| Model | PEFT | Params (%) | Accuracy (↑) | | | | | | | | |
|---|---|---|---|---|---|---|---|---|---|---|---|
| | | | MNLI | SST-2 | MRPC | CoLA | QNLI | QQP | RTE | STS-B | Avg. |
| base | FT | 100% | 87.3 | 94.4 | 87.9 | 62.4 | 92.5 | 91.7 | 78.3 | 90.6 | 85.6 |
| | Adapter* | 0.318% | 87.0 | 93.3 | 88.4 | 60.9 | 92.5 | **90.5** | 76.5 | **90.5** | **85.0** |
| | LoRA* | 0.239% | 86.6 | 93.9 | 88.7 | 59.7 | **92.6** | 90.4 | 75.3 | 90.3 | 84.7 |
| | Adapter$^{FNN}$* | 0.239% | **87.1** | 93.0 | 88.8 | 58.5 | 92.0 | 90.2 | 77.7 | 90.4 | 84.7 |
| | BitFit* | 0.080% | 84.7 | **94.0** | 88.0 | 54.0 | 91.0 | 87.3 | 69.8 | 89.5 | 82.3 |
| | RED* | 0.016% | 83.9 | 93.9 | **89.2** | **61.0** | 90.7 | 87.2 | 78.0 | 90.4 | 84.3 |
| | **DiReFT (ours)** | 0.015% | 82.5 | 92.6 | 88.3 | 58.6 | 91.3 | 86.4 | 76.4 | 89.3 | 83.2 |
| | **LoReFT (ours)** | 0.015% | 83.1 | 93.4 | **89.2** | 60.4 | 91.2 | 87.4 | **79.0** | 90.0 | 84.2 |
| large | FT | 100% | 88.8 | 96.0 | 91.7 | 68.2 | 93.8 | 91.5 | 85.8 | 92.6 | 88.6 |
| | Adapter* | 0.254% | 90.1 | 95.2 | 90.5 | 65.4 | 94.6 | **91.4** | 85.3 | 91.5 | 88.0 |
| | LoRA* | 0.225% | 90.2 | 96.0 | 89.8 | 65.5 | **94.7** | 90.7 | 86.3 | **91.7** | 88.1 |
| | Adapter$^{FNN}$* | 0.225% | **90.3** | 96.1 | **90.5** | 64.4 | 94.3 | 91.3 | 84.8 | 90.2 | 87.7 |
| | RED* | 0.014% | 89.5 | 96.0 | 90.3 | **68.1** | 93.5 | 88.8 | 86.2 | 91.3 | 88.0 |
| | **DiReFT (ours)** | 0.014% | 88.7 | 95.4 | 88.5 | 66.7 | 93.9 | 88.1 | 86.9 | 91.2 | 87.4 |
| | **LoReFT (ours)** | 0.014% | 89.2 | **96.2** | 90.1 | 68.0 | 94.1 | 88.5 | **87.5** | 91.6 | **88.2** |

## 4.5 Natural language understanding

We evaluate LoReFT on the GLUE benchmark [Wang et al., 2018] against existing PEFTs. We use this set of experiments to show LoReFT works well even with small-scale LMs, and can improve representations for classification tasks and not just text generation. We finetune RoBERTa-base (125M) as well as RoBERTa-large (350M) on GLUE, a sequence classification benchmark for natural language understanding (NLU) which covers domains such as sentiment classification and natural language inference. Details about the GLUE benchmark can be found in its original paper. We follow Wu et al. [2024a] for proper evaluation on GLUE validation set: we split the validation set into two sets guarded by a random seed, and we pick the best model with highest in-training validation accuracy to evaluate on the other held-out half for testing accuracy.

**Hyperparameter tuning.** We tune our hyperparameters for each task separately. which is standard for PEFTs. To avoid overfitting to random seeds, we hyperparameter-tune our models with a constant seed, and report averaged results over that and four additional unseen seeds. We describe hyperparameter tuning experiments in Appendix D.1.

**Results.** We report results in table 4. LoReFT obtains comparable performance with PEFT methods on both model sizes when parameter matched with RED, the previous most parameter-efficient PEFT for this task. Furthermore, DiReFT achieves worse performance than most of the PEFTs suggesting LoReFT is a better choice when LM is small. Full results with standard deviation is in table 13. We additionally compare against VeRA [Kopiczko et al., 2024] in appendix D.3.

## 5 Limitations

Due to limited resources, we mainly explored the LLaMA-family of models. In future work, we hope to explore the effectiveness of ReFT on other model families as well as vision–language models such as LLaVA [Liu et al., 2024a]. The capabilities of ReFT have not yet been fully explored due to the large hyperparameter search space; we are interested in automating this search. We provide some initial explorations of LM personalisation with ReFT in a few-shot setting in appendix G.2. We hope to explore why ReFT works, and we provide some of our early explorations focused on memorisation (appendix F.1, appendix F.2). We are also investigating whether learned orthogonal subspaces can be composed together without adaptation. Some encouraging initial findings are in appendix G.1.

## 6 Conclusion

We propose a strong alternative to PEFTs, LoReFT, and we identify an ablation of this method, DiReFT, that trades some performance for increased efficiency. Overall, LoReFT achieves strong per-

formance across benchmarks from four domains while being 15×–65× more efficient than LoRA. Notably, LoReFT establishes new state-of-the-art performance on commonsense reasoning, instruction-following, and natural language understanding against the strongest PEFTs. We also show how our method can be described under a generic framework – ReFT. ReFT is a new approach to finetuning that is more powerful, more parameter-efficient, and more interpretable than any existing PEFTs.

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

# Appendix

## Table of Contents

# A  `pyreft`: A ReFT-native Python Library

To lower the cost of switching from PEFTs to ReFT, we release `pyreft`, a Python library made for training and sharing ReFTs. Our library is built on top of pyvene [Wu et al., 2024b], a library for performing and training activation interventions on arbitrary PyTorch models. We publish our library on PyPI.[6] Any pretrained LM available on HuggingFace is supported through our library `pyreft` for finetuning with ReFT methods, and finetuned models can be easily uploaded to HuggingFace. The following example shows steps to wrap a Llama-2 7B model with a single intervention on the residual stream output of the 19-th layer:

```python
import torch
import transformers
from pyreft import get_reft_model, ReftConfig, LoreftIntervention, ReftTrainerForCausalLM

# loading huggingface model
model_name_or_path = "yahma/llama-7b-hf"
model = transformers.AutoModelForCausalLM.from_pretrained(
    model_name_or_path, torch_dtype=torch.bfloat16, device_map="cuda")
# wrap the model with rank-1 loreft
reft_config = ReftConfig(representations={
    "layer": 19, "component": "block_output",
    "intervention": LoreftIntervention(
    embed_dim=model.config.hidden_size, low_rank_dimension=1)})
reft_model = get_reft_model(model, reft_config)
reft_model.print_trainable_parameters()
```

The wrapped model can be trained for downstream tasks. We also provide data loading helpers to construct training data that is compatible with HuggingFace trainers:

```python
tokenizer = transformers.AutoTokenizer.from_pretrained(model_name_or_path)

# get training data with customised dataloaders
data_module = make_supervised_data_module(
    tokenizer=tokenizer, model=model, layers=[19],
    training_args=training_args, data_args=data_args)
# train
trainer = reft.ReftTrainerForCausalLM(
    model=reft_model, tokenizer=tokenizer, args=training_args, **data_module)
trainer.train()
trainer.save_model(output_dir=training_args.output_dir)
```

# B  Describing existing methods under the ReFT framework

To show the expressivity of the ReFT framework, we cast existing representing-editing methods in the literature into ReFTs.

**General comments about expressivity of ReFT.** Given that previous works have unified PEFTs under a single framework [He et al., 2022a], one may ask **why not express ReFT as a PEFT method?** The main reason is that PEFT frameworks lack the notion of *time* or *sequence* (see the unified PEFT view provided in Table 1 on pg. 5 of He et al., 2022a). In PEFTs, representation modifications are necessarily applied to *every* token in the sequence, even in recent variants such as AdaLoRA [Zhang et al., 2023]. A key aspect of ReFT is that it leverages representations over time and intervenes only on a small number of them while being effective. More importantly, the notation of time is important for future versions of ReFT that intervene on representations *schematically* (e.g. intervene on the first token at some early layers and then intervene on the last token at some later layers). The ability to intervene at different layer and position combinations schematically is also supported in our code. Existing PEFT libraries [7] enforce *weight-based* updates without supporting flexible representation-based interventions.

---

[6] `pip install pyreft`
[7] See https://github.com/huggingface/peft.

## B.1 RED

RED [Wu et al., 2024a] is a simple representation-editing method that applies an element-wise scaling transform $\mathbf{s} \in \mathbb{R}^n$ and adds a bias $\mathbf{b} \in \mathbb{R}^n$ to the hidden representation in every layer. The same intervention is applied to every position (including at generated tokens, increasing inference burden) but separate interventions are learned at each layer. In the ReFT framework, RED is defined as

$$\Phi_{\mathsf{RED}}(\mathbf{h}) = \mathbf{s} \times \mathbf{h} + \mathbf{b} \tag{8}$$

$$\mathbf{I}_{\mathsf{RED}} = \{\langle \Phi_{\mathsf{RED}}, \{1, \ldots, n\}, l\rangle \mid l \in \{1, \ldots, m\}\} \tag{9}$$

The parameters $\phi_{\mathsf{RED}} = \{\mathbf{s}, \mathbf{b}\}$ are learned with gradient descent to minimise a loss function such as language-modelling loss or a classification loss, as in our experiments with LoReFT. We believe that RED is better classified as a kind of adapter due to its application at all positions.

## B.2 Activation addition

Activation addition [Turner et al., 2023] takes the difference in activations at at some positions $p$ and $q$ and layer $l$ given two contrastive prompts $\mathbf{x}^+$ and $\mathbf{x}^-$ as input. It then adds this difference vector, scaled by a tuned constant $c$, to representations at all positions in layer $l$ for some new prompt.

$$\mathbf{a} = \mathbf{h}(\mathbf{x}^+)_p^{(l)} - \mathbf{h}(\mathbf{x}^-)_q^{(l)} \tag{10}$$

$$\Phi_{\mathsf{ActAdd}}(\mathbf{h}) = \mathbf{h} + c \cdot \mathbf{a} \tag{11}$$

$$\mathbf{I}_{\mathsf{ActAdd}} = \{\langle \phi_{\mathsf{ActAdd}}, \{1, \ldots, n\}, l\rangle\} \tag{12}$$

## B.3 RepE

Zou et al. [2023] introduce several intervention methods for controlling model behaviour, which they term *representation engineering*.

First, given a set of prompts $\{\mathbf{x}_1, \ldots, \mathbf{x}_n\}$ designed to elicit the presence of a concept, we randomly pair them, take the difference in activations for each pair, and find the first principle component of the difference vectors at the last token position in some layer of interest $l$ to obtain a *reading vector*:

$$\mathbf{a}_{\mathsf{reading}} = \mathsf{PCA}\left(\left\{\mathbf{h}(\mathbf{x}_i)_{-1}^{(l)} - \mathbf{h}(\mathbf{x}_{i+1})_{-1}^{(l)} \mid i \equiv 0 \bmod 2\right\}\right)_1 \tag{13}$$

One can also used a more structured pairing of constrastive prompts to obtain a *contrast vector*, similar to the difference vector computed in activation addition:

$$\mathbf{a}_{\mathsf{contrast}} = \mathsf{PCA}\left(\left\{\mathbf{h}(\mathbf{x}_i^+)_{-1}^{(l)} - \mathbf{h}(\mathbf{x}_i^-)_{-1}^{(l)} \mid 1 \le i \le n\right\}\right)_1 \tag{14}$$

Then, using either $\mathbf{a}_{\mathsf{reading}}$ or $\mathbf{a}_{\mathsf{contrast}}$, RepE introduces three operators (i.e. parametrisations of $\Phi$) for intervening on activations:

$$\Phi_{\mathsf{RepE,linear}}(\mathbf{h}) = \mathbf{h} \pm c \cdot \mathbf{a} \tag{15}$$

$$\Phi_{\mathsf{RepE,piecewise}}(\mathbf{h}) = \mathbf{h} + c \cdot \mathrm{sign}(\mathbf{a} \cdot \mathbf{h}) \cdot \mathbf{a} \tag{16}$$

$$\Phi_{\mathsf{RepE,projection}}(\mathbf{h}) = \mathbf{h} - c \cdot \frac{\mathbf{a} \cdot \mathbf{h}}{\|\mathbf{a}\|^2} \cdot \mathbf{a} \tag{17}$$

The first two of these are similar to activation addition, while the latter is a scaled one-dimensional distributed interchange intervention that is a special case of LoReFT. These operations are then used to intervene on some set of positions $P \subseteq \{1, \ldots, n\}$ in the layer of interest:

$$\mathbf{I}_{\mathsf{RepE}} = \{\langle \Phi_{\mathsf{RepE}}, P, l\rangle\} \tag{18}$$

RepE introduces another model control method called Low-Rank Representation Adaptation (LoRRA), which is a kind of PEFT rather than a ReFT since it tunes model *weights* using a variant of LoRA.

# C Datasets

## C.1 Commonsense reasoning

We train and evaluate our models on eight datasets covering different domains of open-ended QA tasks:

1. The **BoolQ** [Clark et al., 2019] dataset, which is a question-answering dataset for yes or no naturally occurring questions. We remove the provided passage in the dataset following previous works to ensure a fair comparison.

2. The **PIQA** [Bisk et al., 2020] dataset, which tests physical commonsense reasoning and requires the model to choose one of the provided actions to take based on a hypothesised scenario.

3. The **SIQA** [Sap et al., 2019] dataset, which focus on reasoning about people's actions and their corresponding social consequences.

4. The **HellaSwag** [Zellers et al., 2019] dataset, which asks the model to choose an appropriate ending (or sentence completion) given a context.

5. The **WinoGrande** [Sakaguchi et al., 2021] dataset, inspired by Winograd Schema Challenge [Levesque et al., 2012], asks the model to fill-in-a-blank with binary options given a sentence which requires commonsense reasoning.

6. The ARC Easy set (**ARC-e** [Clark et al., 2018]), which includes genuine grade-school level multiple-choice science questions

7. The ARC Challenge set (**ARC-c**) [Clark et al., 2018]), which is like **ARC-e** but designed in a way that co-occurrence methods are expected to fail to answer correctly.

8. The **OBQA** [Mihaylov et al., 2018] dataset, which is a knowledge-intensive and open-book QA dataset that requires multi-hop reasoning. Dataset statistics and simplified training examples from each dataset are provided in Hu et al. [2023].

Dataset statistics and simplified training examples from each dataset are provided in Hu et al. [2023]. We replicate the experimental setup in Hu et al. [2023] and finetune our models on a combined training dataset (COMMONSENSE170K) of the tasks mentioned above, and evaluate on their individual test set. As in Hu et al. [2023], all of our examples are constructed without the golden or retrieved passages, if provided by the datasets.

## C.2 Arithmetic reasoning

We train and evaluate with seven datasets covering different domains of math world problems:

1. The **AddSub** [Hosseini et al., 2014] dataset, which involves solving arithmetic word problems that include addition and subtraction.

2. The **AQuA** [Ling et al., 2017] dataset, which formulates algebraic word problems as multiple-choice problems.

3. The **GSM8K** [Cobbe et al., 2021] dataset, which consists of grade-school math word problems that require multi-step reasoning.

4. The **MAWPS** [Koncel-Kedziorski et al., 2016] dataset, which contains math word problem with varying complexity.

5. The **MultiArith** [Roy and Roth, 2015] dataset, which contains multi-step arithmetic problems.

6. The **SingleEq** [Koncel-Kedziorski et al., 2015] dataset, which has grade-school math word problems that map to single equations with different length.

7. The **SVAMP** [Patel et al., 2021] dataset, which enhances the original Math World Problem (MWP) challenge by requiring robust reasoning ability that is invariant to structural alternations of the posing problem.

Dataset statistics and simplified training examples from each dataset are provided in Hu et al. [2023]. We replicate the experimental setup in Hu et al. [2023] and finetune our models on a combined training dataset (MATH10K) of four tasks mentioned above: GSM8K, MAWPS, MAWPS-single and AQuA. Different from Hu et al. [2023], selected tasks are excluded for testing since the original paper accidentally leaks testing examples from these tasks into the training set, affecting AddSub, MultiArith and SingleEq. They are included in the MAWPS training dataset, and thus leaked into the training dataset.

## C.3 Natural language understanding

We follow Wu et al. [2024a] for proper evaluation on the GLUE validation set. We split the validation set into two subsets, using one subset guarded by a random seed for in-training evaluation and the other for testing. Specifically, after each training epoch, we evaluate the model on our in-training evaluation set and select the best model across all epochs for testing. For datasets with a large validation set (i.e., QQP, MNLI, and QNLI), we select 1,000 samples for in-training evaluation. For the remaining smaller datasets, we select half of the samples for this purpose. For the evaluation metric, we use the Matthews correlation coefficient for CoLA, the Pearson correlation coefficient for STS-B, and accuracy for the other datasets. For MNLI, we report results only on the matched version.

## D Hyperparameters

### D.1 Hyperparameter tuning and decoding strategy

**Commonsense reasoning and arithmeric reasoning.** We create a standalone development set by taking the last 300 examples from the GSM8K training set. We train our models with the remaining training set of GSM8K and select the hyperparameter settings based on model performance on the development set. We select the hyperparameters using LLaMA-7B, and apply the same settings to LLaMA-13B without additional tuning. We use a maximum sequence length of 512 for training and hyperparameter tuning, and a maximum new token number of 32 for inference. Table 5 and table 6 describes our hyperparameter search space. We use a lower number of epochs (6 instead of 12) for the commonsense reasoning benchmark because the COMMONSENSE170K training set is more than 20 times larger than GSM8K.

During inference, we use greedy decoding without sampling for the commonsense reasoning benchmark, since it is a multi-token classification benchmark, and use the same decoding strategy as in Hu et al. [2023] for the arithmetic reasoning benchmark with a higher temperature 0.3. The reason to switch to a slightly different set of decoding hyperparameters is that the HuggingFace decoding function may throw an error due to statistical instability with close-to-zero probabilities over output tokens with beam search.[8]

**Instruction following.** We finetune LLaMA-7B on Alpaca-52K [Taori et al., 2023] to select hyperparameters. We select the hyperparameter settings based on model performance evaluated with Alpaca-Eval v1.0 [Li et al., 2023], which calculates the win-rate over `text-davinci-003` by using `gpt-4-turbo` as the annotator. We use a maximum sequence length of 768 for training and hyperparameter tuning, and a maximum new token number of 2048 for inference. Table 7 describes our hyperparameter search space.

During inference, we use the same decoding strategy as in RED [Wu et al., 2024a] to ensure a fair comparison. Specifically, we use greedy decoding without sampling, and use a maximum repetition n-gram size of 5 with a repetition penalty of 1.1.

**Natural language understanding.** We conduct hyperparameter tuning with RoBERTa-base and RoBERTa-large for each task individually. We pick the hyperparameters based on testing performance on the held-out validation set with a fixed random seed of 42. We then evaluate our model with additional four unseen seeds {43, 44, 45, 46} for final results. We follow Wu et al. [2024a]'s setting for evaluation. For QQP with RoBERTa-large, there are some stochasticity in runs with the same seed, so we picked the best run out of 3 runs for any particular seed. As reported by Wu et al. [2024a], we also observe that evaluation results on RTE are unstable due to the small size of the dataset. We thus replace several random seeds as in Wu et al. [2024a] to ensure a fair comparison. In addition, we replace one or two random seeds for CoLA for stability. Table 8 describes our hyperparameter search space. Table 9 to table 12 describe our hyperparameter settings for each task. We set the maximum sequence length to 256 for all tasks as in Wu et al. [2024a].

We conduct separate hyperparameter tuning for LoReFT and DiReFT to ensure a fair comparison.

---

[8]See reference ticket: https://github.com/huggingface/transformers/issues/11267.

Table 5: Hyperparameter search space of LLaMA-1 7B models with LoReFT on the GSM8K development set with the best settings underlined. We use greedy decoding without sampling during hyperparameter tuning.

| Hyperparameters | LLaMA-7B w/ GSM8K for LoReFT |
| --- | --- |
| prefix+suffix position $p + s$ | {$p1$+$s1$, $p3$+$s3$, $p5$+$s5$, $p7$+$s7$, $p9$+$s9$, $p11$+$s11$} |
| Tied weight $p$, $s$ | {True, False} |
| Rank $r$ | {8, 16, 32, 64} |
| Layer $L$ (sep. w/ ';') | {0;2;4;6;10;12;14;18, 10;12;14;18;20;22;24;28, 4;6;10;12;14;18;20;22, all} |
| Dropout | {0.00, 0.05} |
| Optimizer | AdamW |
| LR | {$9{\times}10^{-5}$, $1{\times}10^{-4}$, $3{\times}10^{-4}$, $6{\times}10^{-4}$, $9{\times}10^{-4}$, $1{\times}10^{-3}$, $3{\times}10^{-3}$} |
| Weight decay | {0, $1{\times}10^{-3}$, $2{\times}10^{-3}$} |
| LR scheduler | Linear |
| Batch size | {4, 8, 16, 32, 64} |
| Warmup ratio | {0.00, 0.06, 0.10} |
| Epochs | {3, 6, 9, 12, 18} |

Table 6: Hyperparameter search space of LLaMA-1 7B models with DiReFT on the GSM8K development set with the best settings underlined. We use greedy decoding without sampling during hyperparameter tuning.

| Hyperparameters | LLaMA-7B w/ GSM8K for DiReFT |
| --- | --- |
| prefix+suffix position $p + s$ | {$p1$+$s1$, $p3$+$s3$, $p5$+$s5$, $p7$+$s7$, $p9$+$s9$, $p11$+$s11$} |
| Tied weight $p$, $s$ | {True, False} |
| Rank $r$ | {8, 16, 32, 64} |
| Layer $L$ (sep. w/ ';') | {0;2;4;6;10;12;14;18, 10;12;14;18;20;22;24;28, 4;6;10;12;14;18;20;22, all} |
| Dropout | {0.00, 0.05} |
| Optimizer | AdamW |
| LR | {$9{\times}10^{-5}$, $1{\times}10^{-4}$, $3{\times}10^{-4}$, $6{\times}10^{-4}$, $9{\times}10^{-4}$, $1{\times}10^{-3}$, $3{\times}10^{-3}$} |
| Weight decay | {0, $1{\times}10^{-3}$, $2{\times}10^{-3}$, $6{\times}10^{-3}$, $1{\times}10^{-2}$, $2{\times}10^{-2}$, $6{\times}10^{-2}$} |
| LR scheduler | Linear |
| Batch size | {4, 8, 16, 32, 64} |
| Warmup ratio | {0.00, 0.06, 0.10} |
| Epochs | {3, 6, 9, 12, 18} |

Table 7: Hyperparameter search space of LLaMA-1 7B models on Alpaca-52K evaluated by Alpaca-Eval v1.0 with the best settings underlined. We use greedy decoding without sampling during hyperparameter tuning. **LoReFT and DiReFT have the same hyperparameter settings.**

| Hyperparameters | LLaMA-7B w/ Alpaca-52K |
|---|---|
| prefix+suffix position $p + s$ | $\{p1+s1,\ p3+s3,\ \underline{p5+s5},\ p7+s7\}$ |
| Tied weight $p$, $s$ | $\{\underline{\text{True}},\ \text{False}\}$ |
| Rank $r$ | $\{1, 2, 3, \underline{4}, 5, 6\}$ |
| Layer $L$ (sep. w/ ';') | $\{9;18,\ 3;9;18,\ \underline{3;9;18;24}\}$ |
| Dropout | $\{0.00,\ \underline{0.05}\}$ |
| Optimizer | AdamW |
| LR | $9\times10^{-4}$ |
| Weight decay | $0\times10^{-3}$ |
| LR scheduler | Linear |
| Batch size | $\{16, 32, 64, \underline{128}\}$ |
| Warmup ratio | 0.00 |
| Epochs | $\{1, 3, 6, \underline{9}, 12\}$ |

Table 8: Hyperparameter search space of RoBERTa-base and RoBERTa-large models on GLUE evaluated with classification accuracy. Best hyperparameter settings are task-specific, which are specified in separate tables.

| Hyperparameters | RoBERTa-base and RoBERTa-large w/ GLUE |
|---|---|
| prefix+suffix position $p + s$ | $\{p1, p3, p5, p7, p9, p11\}$ |
| Tied weight $p$, $s$ | False |
| Rank $r$ | $\{1, 2\}$ |
| Layer $L$ (sep. w/ ';') | $\{1;3;5;7;9;11,\ \text{all}\}$ |
| Dropout | $\{0.00, 0.05, 0.10, 0.15, 0.20\}$ |
| Optimizer | AdamW |
| LR | $\{1\times10^{-4}, 2\times10^{-4}, 3\times10^{-4}, 4\times10^{-4}, 5\times10^{-4}\}$, $\{6\times10^{-4}, 9\times10^{-4}, 1\times10^{-3}, 3\times10^{-3}\}$ |
| Weight decay | $\{0, 1\times10^{-4}, 6\times10^{-4}, 1\times10^{-3}, 6\times10^{-3}, 1\times10^{-2}, 2\times10^{-2}, 4\times10^{-2}\}$ |
| LR scheduler | Linear |
| Batch size | $\{16, 32, 64, 128\}$ |
| Warmup ratio | $\{0, 5\times10^{-3}, 6\times10^{-3}, 3\times10^{-2}, 5\times10^{-2}, 6\times10^{-2}, 1\times10^{-1}, 2\times10^{-1}\}$ |
| Epochs | $\{20, 30, 40, 50, 60\}$ |

Table 9: Hyperparameter settings of RoBERTa-base models on GLUE for LoReFT.

| Hyperparameters | MNLI | SST-2 | MRPC | CoLA | QNLI | QQP | RTE | STS-B |
|---|---|---|---|---|---|---|---|---|
| position $p$ | $p1$ | $p3$ | $p3$ | $p3$ | $p11$ | $p11$ | $p3$ | $p3$ |
| Tied weight | | | | False | | | | |
| Rank $r$ | | | | 1 | | | | |
| Layer $L$ | | | | all | | | | |
| Dropout | 0.05 | 0.10 | 0.05 | 0.20 | 0.05 | 0.05 | 0.05 | 0.05 |
| Optimizer | | | | AdamW | | | | |
| LR | $6\times10^{-4}$ | $6\times10^{-4}$ | $3\times10^{-4}$ | $4\times10^{-4}$ | $9\times10^{-4}$ | $6\times10^{-4}$ | $9\times10^{-4}$ | $6\times10^{-4}$ |
| Weight decay | | | | 0.00 | | | | |
| LR scheduler | | | | Linear | | | | |
| Batch size | | | | 32 | | | | |
| Warmup ratio | $6\times10^{-2}$ | $1\times10^{-1}$ | 0 | $5\times10^{-3}$ | $1\times10^{-1}$ | 0 | 0 | $3\times10^{-2}$ |
| Epochs | 40 | 40 | 40 | 60 | 20 | 40 | 60 | 60 |

Table 10: Hyperparameter settings of RoBERTa-large models on GLUE for LoReFT.

| Hyperparameters | MNLI | SST-2 | MRPC | CoLA | QNLI | QQP | RTE | STS-B |
|---|---|---|---|---|---|---|---|---|
| position $p$ | $p1$ | $p3$ | $p3$ | $p3$ | $p11$ | $p11$ | $p3$ | $p3$ |
| Tied weight | | | | False | | | | |
| Rank $r$ | | | | 1 | | | | |
| Layer $L$ | | | | all | | | | |
| Dropout | 0.05 | 0.05 | 0.20 | 0.20 | 0.05 | 0.05 | 0.05 | 0.05 |
| Optimizer | | | | AdamW | | | | |
| LR | $6\times10^{-4}$ | $6\times10^{-4}$ | $3\times10^{-4}$ | $1\times10^{-4}$ | $9\times10^{-4}$ | $6\times10^{-4}$ | $6\times10^{-4}$ | $8\times10^{-4}$ |
| Weight decay | | | | 0.00 | | | | |
| LR scheduler | | | | Linear | | | | |
| Batch size | | | | 32 | | | | |
| Warmup ratio | 0.00 | 0.10 | 0.06 | 0.20 | 0.10 | 0.06 | 0.00 | 0.20 |
| Epochs | 20 | 20 | 30 | 30 | 20 | 20 | 30 | 30 |

Table 11: Hyperparameter settings of RoBERTa-base models on GLUE for DiReFT.

| Hyperparameters | MNLI | SST-2 | MRPC | CoLA | QNLI | QQP | RTE | STS-B |
|---|---|---|---|---|---|---|---|---|
| position $p$ | $p1$ | $p3$ | $p5$ | $p1$ | $p11$ | $p11$ | $p1$ | $p3$ |
| Tied weight | | | | False | | | | |
| Rank $r$ | | | | 1 | | | | |
| Layer $L$ | | | | all | | | | |
| Dropout | 0.05 | 0.10 | 0.05 | 0.00 | 0.05 | 0.05 | 0.00 | 0.05 |
| Optimizer | | | | AdamW | | | | |
| LR | $6\times10^{-4}$ | $6\times10^{-4}$ | $3\times10^{-4}$ | $6\times10^{-4}$ | $9\times10^{-4}$ | $6\times10^{-4}$ | $9\times10^{-4}$ | $6\times10^{-4}$ |
| Weight decay | 0.00 | 0.00 | 0.00 | 0.04 | 0.00 | 0.00 | 0.04 | 0.00 |
| LR scheduler | | | | Linear | | | | |
| Batch size | 32 | 32 | 32 | 32 | 32 | 32 | 8 | 32 |
| Warmup ratio | $6\times10^{-2}$ | $1\times10^{-1}$ | $1\times10^{-1}$ | 0 | $1\times10^{-1}$ | 0 | 0 | $3\times10^{-2}$ |
| Epochs | 40 | 40 | 40 | 60 | 20 | 40 | 60 | 60 |

Table 12: Hyperparameter settings of RoBERTa-large models on GLUE for DiReFT.

| Hyperparameters | MNLI | SST-2 | MRPC | CoLA | QNLI | QQP | RTE | STS-B |
|---|---|---|---|---|---|---|---|---|
| position $p$ | $p1$ | $p3$ | $p1$ | $p1$ | $p11$ | $p7$ | $p3$ | $p3$ |
| Tied weight | | | | False | | | | |
| Rank $r$ | | | | 1 | | | | |
| Layer $L$ | | | | all | | | | |
| Dropout | 0.05 | 0.05 | 0.10 | 0.15 | 0.05 | 0.05 | 0.05 | 0.05 |
| Optimizer | | | | AdamW | | | | |
| LR | $6\times10^{-4}$ | $6\times10^{-4}$ | $9\times10^{-4}$ | $9\times10^{-4}$ | $9\times10^{-4}$ | $9\times10^{-4}$ | $6\times10^{-4}$ | $8\times10^{-4}$ |
| Weight decay | 0 | 0 | 0 | 0 | 0 | 0 | $6\times10^{-3}$ | 0 |
| LR scheduler | | | | Linear | | | | |
| Batch size | | | | 32 | | | | |
| Warmup ratio | 0.00 | 0.10 | 0.00 | 0.00 | 0.10 | 0.10 | 0.00 | 0.10 |
| Epochs | 20 | 20 | 50 | 60 | 20 | 20 | 30 | 30 |

Table 13: Accuracy comparison of RoBERTa-base and RoBERTa-large against existing PEFT methods on the GLUE benchmark with **standard deviation (SD)**. *Performance results of all baseline methods are taken from Wu et al. [2024a]. We report averaged performance of five runs with distinct random seeds for our method. Param. (%) is calculated by dividing the number of trainable parameters (excluding the number of parameters of the classification head) with the number of parameter of the base LM.

| Model | PEFT | Params (%) | Accuracy (↑) (SD) | | | | | | | | |
| --- | --- | --- | --- | --- | --- | --- | --- | --- | --- | --- | --- |
| | | | **MNLI** | **SST-2** | **MRPC** | **CoLA** | **QNLI** | **QQP** | **RTE** | **STS-B** | **Avg.** |
| base | FT | 100% | $87.3_{(0.34)}$ | $94.4_{(0.96)}$ | $87.9_{(0.91)}$ | $62.4_{(3.29)}$ | $92.5_{(0.22)}$ | $91.7_{(0.19)}$ | $78.3_{(3.20)}$ | $90.6_{(0.59)}$ | 85.6 |
| | Adapter* | 0.318% | $87.0_{(0.28)}$ | $93.3_{(0.40)}$ | $88.4_{(1.54)}$ | $60.9_{(3.09)}$ | $92.5_{(0.02)}$ | $\mathbf{90.5}_{(0.08)}$ | $76.5_{(2.26)}$ | $\mathbf{90.5}_{(0.35)}$ | **85.0** |
| | LoRA* | 0.239% | $86.6_{(0.23)}$ | $93.9_{(0.49)}$ | $88.7_{(0.76)}$ | $59.7_{(4.36)}$ | $\mathbf{92.6}_{(0.10)}$ | $90.4_{(0.08)}$ | $75.3_{(2.79)}$ | $90.3_{(0.54)}$ | 84.7 |
| | Adapter^FNN* | 0.239% | $\mathbf{87.1}_{(0.10)}$ | $93.0_{(0.05)}$ | $88.8_{(1.38)}$ | $58.5_{(1.69)}$ | $92.0_{(0.28)}$ | $90.2_{(0.07)}$ | $77.7_{(1.93)}$ | $90.4_{(0.31)}$ | 84.7 |
| | BitFit* | 0.080% | $84.7_{(0.08)}$ | $\mathbf{94.0}_{(0.87)}$ | $88.1_{(1.57)}$ | $54.0_{(3.07)}$ | $91.0_{(0.05)}$ | $87.3_{(0.02)}$ | $69.8_{(1.51)}$ | $89.5_{(0.35)}$ | 82.3 |
| | RED* | 0.016% | $83.9_{(0.14)}$ | $93.9_{(0.31)}$ | $\mathbf{89.2}_{(0.98)}$ | $\mathbf{61.0}_{(2.96)}$ | $90.7_{(0.35)}$ | $87.2_{(0.17)}$ | $78.0_{(2.06)}$ | $90.4_{(0.32)}$ | 84.3 |
| | **DiReFT (ours)** | 0.015% | $82.5_{(0.22)}$ | $92.6_{(0.76)}$ | $88.3_{(1.23)}$ | $58.6_{(1.99)}$ | $91.3_{(0.19)}$ | $86.4_{(0.27)}$ | $76.4_{(1.48)}$ | $89.3_{(0.56)}$ | 83.2 |
| | **LoReFT (ours)** | 0.015% | $83.1_{(0.26)}$ | $93.4_{(0.64)}$ | $\mathbf{89.2}_{(2.62)}$ | $60.4_{(2.60)}$ | $91.2_{(0.25)}$ | $87.4_{(0.23)}$ | $\mathbf{79.0}_{(2.76)}$ | $90.0_{(0.29)}$ | 84.2 |
| large | FT | 100% | $88.8_{(0.45)}$ | $96.0_{(0.66)}$ | $91.7_{(1.73)}$ | $68.2_{(2.62)}$ | $93.8_{(0.33)}$ | $91.5_{(1.28)}$ | $85.8_{(1.40)}$ | $92.6_{(0.16)}$ | 88.6 |
| | Adapter* | 0.254% | $90.1_{(0.12)}$ | $95.2_{(0.48)}$ | $90.5_{(0.59)}$ | $65.4_{(2.24)}$ | $94.6_{(0.17)}$ | $\mathbf{91.4}_{(0.13)}$ | $85.3_{(1.34)}$ | $91.5_{(0.33)}$ | 88.0 |
| | LoRA* | 0.225% | $90.2_{(0.25)}$ | $96.0_{(0.85)}$ | $89.8_{(2.09)}$ | $65.5_{(2.02)}$ | $\mathbf{94.7}_{(0.21)}$ | $90.7_{(0.91)}$ | $86.3_{(2.41)}$ | $\mathbf{91.7}_{(0.44)}$ | 88.1 |
| | Adapter^FNN* | 0.225% | $\mathbf{90.3}_{(0.15)}$ | $96.1_{(0.75)}$ | $\mathbf{90.5}_{(1.26)}$ | $64.4_{(1.56)}$ | $94.3_{(0.39)}$ | $91.3_{(0.24)}$ | $84.8_{(2.01)}$ | $90.2_{(0.24)}$ | 87.7 |
| | RED* | 0.014% | $89.5_{(0.38)}$ | $96.0_{(0.48)}$ | $90.3_{(1.40)}$ | $\mathbf{68.1}_{(1.69)}$ | $93.5_{(0.33)}$ | $88.8_{(0.11)}$ | $86.2_{(1.40)}$ | $91.3_{(0.21)}$ | 88.0 |
| | **DiReFT (ours)** | 0.014% | $88.7_{(0.13)}$ | $95.4_{(0.60)}$ | $88.5_{(2.16)}$ | $66.7_{(2.21)}$ | $93.9_{(0.39)}$ | $88.1_{(0.47)}$ | $86.9_{(1.56)}$ | $91.2_{(0.29)}$ | 87.4 |
| | **LoReFT (ours)** | 0.014% | $89.2_{(0.27)}$ | $\mathbf{96.2}_{(0.72)}$ | $90.1_{(1.17)}$ | $68.0_{(1.44)}$ | $94.1_{(0.35)}$ | $88.5_{(0.45)}$ | $\mathbf{87.5}_{(1.49)}$ | $91.6_{(0.43)}$ | **88.2** |

### D.2 Suggestions on choosing hyperparameters for ReFT

Similar to PEFTs or finetuning, ReFT can be sensitive to hyperparameter settings. Here, we recommand a non-exhaustive list for choosing the best hyperparameter settings for your tasks:

- **Intervening on multiple positions delivers significant gains**. We find that intervening only on a single token position (e.g., just the first one or the last one) is always less optimal than intervening on multiple tokens. However, intervening on excessive number of tokens might harm performance by slowing down convergence.

- **Intervening on all layers first, and then shrink down**. Intervening on all layers often provides a good baseline. We recommand users to start with all layers, and shrink down the number of intervening layers depending on the desired performance–parameter count balance.

- **Higher rank may not entail better performance**. High rank entails higher parameter count, but it does not always bring performance gain (likely due to slower convergence). We recommend users to start with a rank that is lower than 32 (e.g. rank 4).

- **Tie intervention weights as much as you can**. In the paper, we explore tying the intervention weights between prefix and suffix token positions. It automatically halves the parameter count, and it can result in better performance as well. We suspect weight sharing across layers may also help.

- **Hyperparameter tuning with learning rate, warmup ratio, dropout rate and weight decay should go after other hyperparameters**. These classic neural-network training hyperparameters can play a role, yet they have much smaller effect than previous ones.

### D.3 Additional hyperparameter-tuning results of LoReFT

As a result of our hyperparameter searching process, LoReFT is trained with more epochs compared to LoRA [Hu et al., 2022] or DoRA [Liu et al., 2024c]. This raises the concern whether our performance gain is purely due to the larger number of epochs. We thus rerun our experiments with the exact same number of epochs and effective batch size as LoRA or DoRA. Results are shown in table 14 and table 15. With matched hyperparameters, LoReFT shows similar results by outperforming previous methods significantly on eight commonsense reasoning datasets.

Recently, VeRA was proposed as a new variant of LoRA that further reduces the number of trainable parameters while maintaining performance [Kopiczko et al., 2024]. Table 16 shows our results compared against VeRA as well as the baseline numbers reported in VeRA's paper. We include this set of results in the appendix, given that the hyperparameter tuning process is drastically different from ours.[9] The original VeRA implementation records the performance of the best epoch on the validation set, which could cause overfitting since results are selected based on test set performance.

---

[9]VeRA's original implementation can be found at `https://openreview.net/notes/edits/attachment?id=D0dcbrnPq0&name=supplementary_material`.

Table 14: Accuracy comparison of LLaMA-7B and LLaMA-13B against existing PEFT methods on eight commonsense reasoning datasets. *Performance results of all baseline methods are taken from Liu et al. [2024c]. We report averaged performance of three runs with distinct random seeds for our method. For LoReFT, Param. (%) is calculated by dividing the number of trainable parameters by the number of parameters of the base LM. We include **LoReFT**$_{e=3}$, which is trained with 3 epochs — the same number of epochs as DoRA, but with a reduced batch size of 16 to ensure an equivalent number of gradient sets.

| Model | PEFT | Params (%) | Accuracy (↑) | | | | | | | | |
|---|---|---|---|---|---|---|---|---|---|---|---|
| | | | BoolQ | PIQA | SIQA | HellaS. | WinoG. | ARC-e | ARC-c | OBQA | Avg. |
| ChatGPT* | — | — | 73.1 | 85.4 | 68.5 | 78.5 | 66.1 | 89.8 | 79.9 | 74.8 | 77.0 |
| LLaMA-7B | PrefT* | 0.039% | 64.3 | 76.8 | 73.9 | 42.1 | 72.1 | 72.9 | 54.0 | 60.6 | 64.6 |
| | Adapter$^S$* | 1.953% | 63.0 | 79.2 | 76.3 | 67.9 | 75.7 | 74.5 | 57.1 | 72.4 | 70.8 |
| | Adapter$^P$* | 3.542% | 67.9 | 76.4 | 78.8 | 69.8 | 78.9 | 73.7 | 57.3 | 75.2 | 72.3 |
| | LoRA* | 0.826% | 68.9 | 80.7 | 77.4 | 78.1 | 78.8 | 77.8 | 61.3 | 74.8 | 74.7 |
| | DoRA (half)* | 0.427% | **70.0** | 82.6 | 79.7 | 83.2 | 80.6 | 80.6 | 65.4 | 77.6 | 77.5 |
| | DoRA* | 0.838% | 68.5 | 82.9 | 79.6 | 84.8 | 80.8 | 81.4 | 65.8 | **81.0** | 78.1 |
| | **LoReFT**$_{e=3}$ | 0.031% | 68.3 | 83.5 | 79.7 | 92.7 | 82.6 | 83.2 | 67.4 | 78.5 | 79.5 |
| | **LoReFT (ours)** | 0.031% | 69.3 | **84.4** | **80.3** | **93.1** | **84.2** | 83.2 | **68.2** | 78.9 | **80.2** |
| LLaMA-13B | PrefT* | 0.031% | 65.3 | 75.4 | 72.1 | 55.2 | 68.6 | 79.5 | 62.9 | 68.0 | 68.4 |
| | Adapter$^S$* | 1.586% | 71.8 | 83.0 | 79.2 | 88.1 | 82.4 | 82.5 | 67.3 | 81.8 | 79.5 |
| | Adapter$^P$* | 2.894% | **72.5** | 84.9 | 79.8 | 92.1 | 84.7 | 84.2 | 71.2 | 82.4 | 81.5 |
| | LoRA* | 0.670% | 72.1 | 83.5 | 80.5 | 90.5 | 83.7 | 82.8 | 68.3 | 82.4 | 80.5 |
| | DoRA (half)* | 0.347% | **72.5** | 85.3 | 79.9 | 90.1 | 82.9 | 82.7 | 69.7 | 83.6 | 80.8 |
| | DoRA* | 0.681% | 72.4 | 84.9 | 81.5 | 92.4 | 84.2 | 84.2 | 69.6 | 82.8 | 81.5 |
| | **LoReFT**$_{e=3}$ | 0.025% | 72.0 | 85.6 | 82.1 | 94.8 | 85.3 | 86.9 | 73.0 | 85.0 | 83.1 |
| | **LoReFT (ours)** | 0.025% | 72.1 | **86.3** | 81.8 | **95.1** | **87.2** | 86.2 | 73.7 | 84.2 | **83.3** |

Table 15: Accuracy comparison of LLaMA-7B and LLaMA-13B against existing PEFT methods on four arithmetic reasoning datasets. *Performance results of all baseline methods are taken from Hu et al. [2023]. We report averaged performance of three runs with distinct random seeds for our method. We include **LoReFT**$_{e=3}$, which is trained with 3 epochs — the same number of epoch as DoRA, but with a reduced batch size of 16 to ensure an equivalent number of gradient sets.

| Model | PEFT | Params (%) | Accuracy (↑) | | | | |
|---|---|---|---|---|---|---|---|
| | | | AQuA | GSM8K | MAWPS | SVAMP | Avg. |
| LLaMA-7B | PrefT* | 0.039% | 14.2 | 24.4 | 63.4 | 38.1 | 35.0 |
| | Adapter$^S$* | 1.953% | 15.0 | 33.3 | 77.7 | **52.3** | 44.6 |
| | Adapter$^P$* | 3.542% | 18.1 | 35.3 | **82.4** | 49.6 | 46.4 |
| | LoRA* | 0.826% | 18.9 | **37.5** | 79.0 | 52.1 | **46.9** |
| | **LoReFT**$_{e=3}$ | 0.031% | 22.4 | 21.6 | 69.5 | 43.6 | 39.3 |
| | **LoReFT (ours)** | 0.031% | **21.4** | 26.0 | 76.2 | 46.8 | 42.6 |
| LLaMA-13B | PrefT* | 0.031% | 15.7 | 31.1 | 66.8 | 41.4 | 38.8 |
| | Adapter$^S$* | 1.586% | 22.0 | 44.0 | 78.6 | 50.8 | 48.9 |
| | Adapter$^P$* | 2.894% | 20.5 | 43.3 | 81.1 | **55.7** | 50.2 |
| | LoRA* | 0.670% | 18.5 | **47.5** | **83.6** | 54.6 | **51.1** |
| | **LoReFT**$_{e=3}$ | 0.025% | 23.4 | 35.5 | 81.8 | 54.6 | 48.8 |
| | **LoReFT (ours)** | 0.025% | **23.6** | 38.1 | 82.4 | 54.2 | 49.6 |

Table 16: Accuracy comparison of RoBERTa-base and RoBERTa-large against existing PEFT methods on the GLUE benchmark. *Performance results of all baseline methods are taken from Kopiczko et al. [2024]. To ensure a fair comparison, we report **median performance** of five runs with distinct random seeds for our method.

| Model | PEFT | Params (%) | Accuracy (↑) | | | | | | |
|---|---|---|---|---|---|---|---|---|---|
| | | | **SST-2** | **MRPC** | **CoLA** | **QNLI** | **RTE** | **STS-B** | **Avg.** |
| base | FT | 100% | 94.8 | 90.2 | 63.6 | 92.8 | 78.7 | 91.2 | 85.2 |
| | BitFit | 0.080% | 93.7 | **92.7** | 62.0 | 91.8 | 81.5 | 90.8 | 85.4 |
| | Adpt$^D$ | 0.239% | 94.2 | 88.5 | 60.8 | 93.1 | 71.5 | 89.7 | 83.0 |
| | Adpt$^D$ | 0.717% | 94.7 | 88.4 | 62.6 | 93.0 | 75.9 | 90.3 | 84.2 |
| | LoRA | 0.239% | **95.1** | 89.7 | 63.4 | **93.3** | **86.6** | **91.5** | **86.6** |
| | VeRA | 0.034% | 94.6 | 89.5 | **65.6** | 91.8 | 78.8 | 90.7 | 85.2 |
| | **DiReFT (ours)** | 0.015% | 92.2 | 88.7 | 59.5 | 91.3 | 77.0 | 89.6 | 83.0 |
| | **LoReFT (ours)** | 0.015% | 93.6 | 87.8 | 59.1 | 91.3 | 79.9 | 90.0 | 83.6 |
| large | Adpt$^P$ | 0.845% | 96.1 | 90.2 | **68.3** | **94.8** | 83.8 | 92.1 | 87.6 |
| | Adpt$^P$ | 0.225% | **96.6** | 89.7 | 67.8 | **94.8** | 80.1 | 91.9 | 86.8 |
| | Adpt$^H$ | 1.690% | 96.2 | 88.7 | 66.5 | 94.7 | 83.4 | 91.0 | 86.8 |
| | Adpt$^H$ | 0.225% | 96.3 | 87.7 | 66.3 | 94.7 | 72.9 | 91.5 | 84.9 |
| | LoRA-FA | 1.042% | 96.0 | 90.0 | 68.0 | 94.4 | 86.1 | 92.0 | 87.8 |
| | LoRA | 0.225% | 96.2 | 90.2 | 68.2 | 94.8 | 85.2 | **92.3** | 87.8 |
| | VeRA | 0.017% | 96.1 | **90.9** | 68.0 | 94.4 | 85.9 | 91.7 | 87.8 |
| | **DiReFT (ours)** | 0.014% | 95.2 | 88.2 | 66.7 | 94.0 | 86.3 | 91.0 | 86.9 |
| | **LoReFT (ours)** | 0.014% | 96.1 | 90.2 | 68.2 | 94.1 | **87.8** | 91.5 | **88.0** |

Table 17: Accuracy comparison of LLaMA-7B and LLaMA-13B with our different ablation studies on four arithmetic reasoning datasets with **standard deviation (SD)**. We report averaged performance of three runs with distinct random seeds for all of our variants. **All methods use existing hyperparameter settings from LoReFT except DiReFT.**

| Model | $\Phi(\mathbf{h})$ | Params (%) | Accuracy (↑) | | | | |
|---|---|---|---|---|---|---|---|
| | | | AQuA | GSM8K | MAWPS | SVAMP | Avg. |
| LLaMA-7B | $\mathbf{h} + \mathbf{R}^\top \mathbf{b}$ | 0.016% | 14.4 | 14.2 | 59.9 | 36.8 | $31.3_{(0.47)}$ |
| | $\mathbf{h} + \mathbf{R}^\top (\mathbf{b} - \mathbf{Rh})$ | 0.016% | 20.1 | 21.2 | 67.9 | 39.2 | $37.1_{(0.19)}$ |
| | $\mathbf{h} + \mathbf{R}^\top (\mathbf{Wh} + \mathbf{b})$ | 0.031% | 21.3 | 27.4 | 76.6 | 46.3 | $42.9_{(0.37)}$ |
| | $\mathbf{h} + \mathbf{W}_2^\top (\mathbf{W}_1\mathbf{h} + \mathbf{b} - \mathbf{W}_2\mathbf{h})$ | 0.031% | 23.1 | 25.5 | 75.4 | 45.6 | $42.4_{(0.71)}$ |
| | **DiReFT** | 0.031% | 21.3 | 24.1 | 74.5 | 42.7 | $40.6_{(0.44)}$ |
| | **LoReFT** | 0.031% | 21.4 | 26.0 | 76.2 | 46.8 | $42.6_{(0.46)}$ |
| LLaMA-13B | $\mathbf{h} + \mathbf{R}^\top \mathbf{b}$ | 0.013% | 16.8 | 25.3 | 69.3 | 46.8 | $39.5_{(0.81)}$ |
| | $\mathbf{h} + \mathbf{R}^\top (\mathbf{b} - \mathbf{Rh})$ | 0.013% | 21.9 | 35.6 | 80.3 | 51.7 | $47.4_{(0.64)}$ |
| | $\mathbf{h} + \mathbf{R}^\top (\mathbf{Wh} + \mathbf{b})$ | 0.025% | 25.1 | 36.7 | 81.9 | 53.6 | $49.3_{(0.39)}$ |
| | $\mathbf{h} + \mathbf{W}_2^\top (\mathbf{W}_1\mathbf{h} + \mathbf{b} - \mathbf{W}_2\mathbf{h})$ | 0.025% | 23.5 | 36.5 | 82.1 | 54.1 | $49.0_{(0.63)}$ |
| | **DiReFT** | 0.025% | 20.5 | 35.8 | 80.8 | 54.8 | $48.0_{(1.23)}$ |
| | **LoReFT** | 0.025% | 23.6 | 38.1 | 82.4 | 54.2 | $49.6_{(0.71)}$ |

# E   Ablating the parametrisation of LoReFT

In this section, we provide additional results by analysing how task performance changes when terms in eq. (2) are ablated. We reevaluate LLaMA-1 7B and 13B with the same set of hyperparameters on the arithmetic reasoning benchmark using variants of the LoReFT intervention function $\Phi$. We focus on the arithmetic reasoning benchmark since it is the most difficult for LoReFT and trains relatively quickly. We conduct experiments with the following parametrisations:

1. $\Phi(\mathbf{h}) = \mathbf{h} + \mathbf{W}_2^\top (\mathbf{W}_1\mathbf{h} + \mathbf{b} - \mathbf{W}_2\mathbf{h})$ where both $\mathbf{W}_1, \mathbf{W}_2 \in \mathbb{R}^{r \times d}$ are low-rank Non-orthogonal linear projection matrices. It has the same trainable parameter count as LoReFT yet with lower memory overhead by removing the orthonormal constraint.

2. $\Phi(\mathbf{h}) = \mathbf{h} + \mathbf{R}^\top (\mathbf{Wh} + \mathbf{b})$ which directly edits the representation in a learned linear subspace. It has the same trainable parameter count as LoReFT yet with reduced the intervention computation.

3. $\Phi(\mathbf{h}) = \mathbf{h} + \mathbf{R}^\top (\mathbf{b} - \mathbf{Rh})$ which makes the linear subspace intervention a constant bias term that is input-independent. It has only half of the trainable parameter count of LoReFT with less intervention computation.

4. $\Phi(\mathbf{h}) = \mathbf{h} + \mathbf{R}^\top \mathbf{b}$. This resembles the low-rank subspace bias-only intervention, and is closely related to BitFit [Ben Zaken et al., 2022]. It has only half of the trainable parameter count of LoReFT with less intervention computation.

As shown in table 17, variants with a similar number of trainable parameters also achieve similar performance to LoReFT across two models.

# F    Memorisation experiments

## F.1    A single vector is worth a thousand tokens

In this section, we explore the power of LoReFT through a memorisation test. Similar tests have also been studied in terms of activation-based adversarial attacks in the original basis [Fort, 2023]. Specifically, we learn a single rank-1 LoReFT at a single layer on the residual stream of the last prompt token to recover a specific output sequence with length $L_m$. For simplicity, we simplify LoReFT in Eqn. 2 by removing $\mathbf{W}\mathbf{h}$ to make the intervention input-independent, where we learn a single scalar $\mathbf{b}$ besides the low-rank matrix. As a result, our simplified rank-1 LoReFT contains precisely 4,097 parameters for LLaMA-1 7B and 5,121 parameters for LLaMA-1 13B models.[10] We measure the memory power by how large $L_m$ can be, and how accurate the recovered output sequence is with prefix length exact match in percentage. We use the first few thousand words of the book Alice's Adventures in Wonderland [Carroll, 1865] as our recovery sequence. Our prompt is constructed as `ALIC#ID1->` followed by model generations. We train with 1000 epochs with a learning rate of $4 \times 10^{-3}$ and a linear learning rate scheduler without warm-up.

As shown in fig. 3 and fig. 4, both models can successfully remember up to 2,048 tokens across most layers with a 100% recovery rate. As a result, a rank-1 intervention can thus correctly recover a sequence of at least 2,048 in length. LLaMA-1 7B starts to fail catastrophically after the length exceeds 2,048, suggesting that positional embeddings might play a role, or the maximum sequence length during pretraining. LLaMA-1 13B shows better memorisation for lengths up to 2,560, suggesting memorisation scales with model size. Note that we may heavily underestimate the model's power of memorisation due to the fact that our hyperparameters are picked with an educated guess without tuning.

From fig. 5 to fig. 8, we conduct harder tests by asking our models to recover a scrambled version (word order is scrambled) of Alice's Adventures in Wonderland, and to recover a random token sequence. Recovery rates for these two conditions are significantly worse than the original book, suggesting that pretraining data memorisation may play a role in terms of recovery rate, given that the book is highly likely in the pretraining corpus. Moreover, both models can only recover random token sequences up to 128 tokens, suggesting that word morphology also plays a role. Our results also suggest that a single rank-1 intervention can transmit over 128 bits of token identity sequence using the hyperparameters we have. [11]

---

[10]These parameters take about 17.5KB of disk space.
[11]Our code is at https://github.com/stanfordnlp/pyreft/tree/main/examples/memorisation.

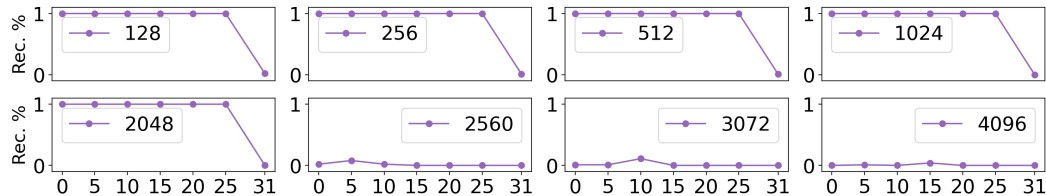

Figure 3: Memorisation test results for **LLaMA-1 7B model** on recovering first n-th tokens of the Alice's Adventures in Wonderland by rank-1 LoReFT intervention on various layers of the last token's residual stream. Rec. % is measured by the percentage of prefix matches.

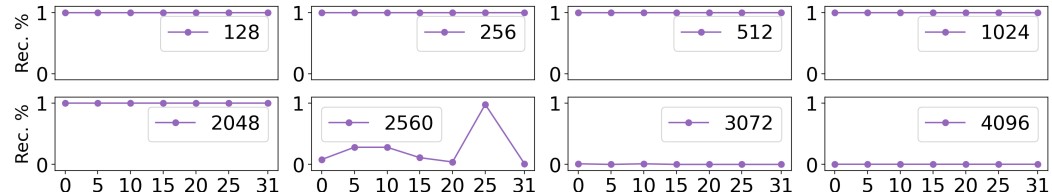

Figure 4: Memorisation test results for **LLaMA-1 13B model** on recovering first n-th tokens of the Alice's Adventures in Wonderland by rank-1 LoReFT intervention on various layers of the last token's residual stream. Rec. % is measured by the percentage of prefix matches.

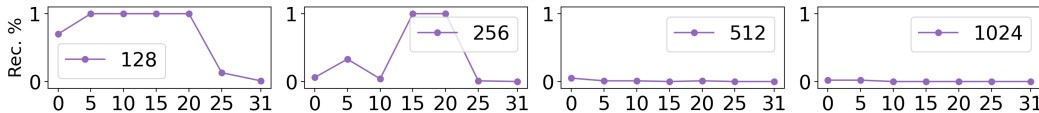

Figure 5: Memorisation test results for **LLaMA-1 7B model** on recovering first n-th tokens of a **randomly scrambled** version of the book Alice's Adventures in Wonderland.

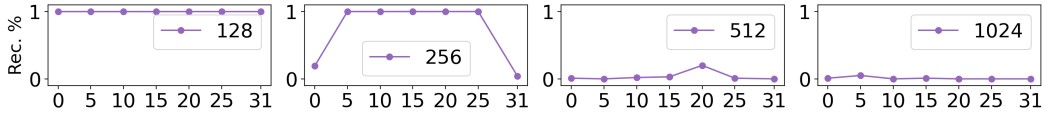

Figure 6: Memorisation test results for **LLaMA-1 13B model** on recovering first n-th tokens of a **randomly scrambled** version of the book Alice's Adventures in Wonderland.

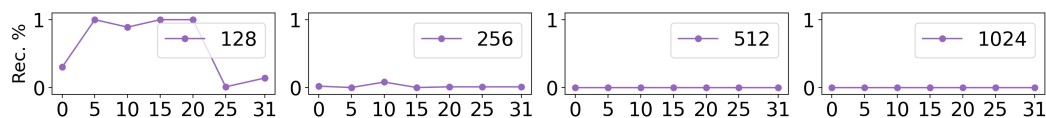

Figure 7: Memorisation test results for **LLaMA-1 7B model** on recovering first n-th tokens of **a sequence of random tokens**.

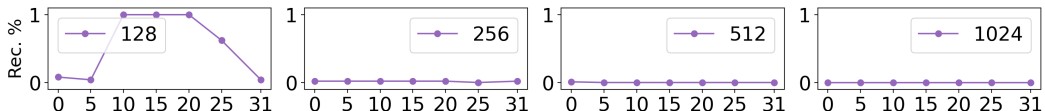

Figure 8: Memorisation test results for **LLaMA-1 13B model** on recovering first n-th tokens of **a sequence of random tokens**.

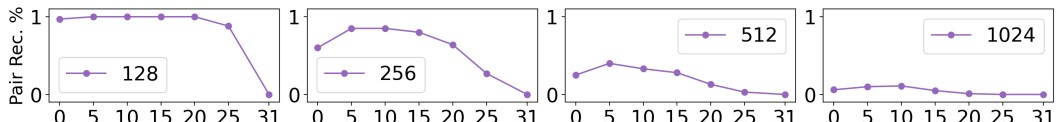

Figure 9: Multitude test results for **LLaMA-1 7B model** on recovering $n$ input-output pairs where each pair constitutes an input prompt as RAND#ID1-> with varying IDs and a single random token output.

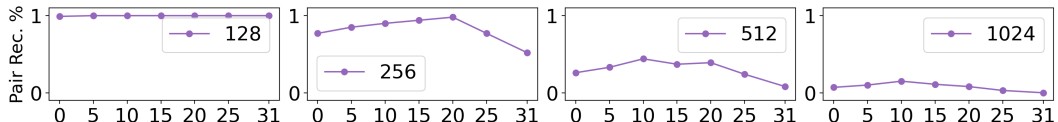

Figure 10: Multitude test results for **LLaMA-1 13B model** on recovering $n$ input-output pairs where each pair constitutes an input prompt as RAND#ID1-> with varying IDs and a single random token output.

## F.2   A single vector can memorise a codebook with 256 entries

Our memorisation tests in appendix F.1 test how long of a sequence we can encode in a rank-1 intervention. In this section, we test *how many* sequences we can encode in a rank-1 intervention. Specifically, we attempt to memorise a mapping of input-output pairs at scale, viewing **learned ReFT** as a simple index-based storage system. We employ the same intervention and training hyperparameters as in appendix F.1, but with a different training dataset. Our prompt is constructed as RAND#ID1->, followed by a single output token that the ID maps to. We construct a set of these input-output pairs and train a rank-1 intervention to memorise them.

We present our results in fig. 9 and fig. 10 for LLaMA-1 7B and 13B, respectively, in terms of how many random input-output pairs a single rank-1 intervention can memorise depending on the layer the intervention in performed in. Our results suggest that a rank-1 intervention can reliably remember up to 256 pairs, with near-perfect recall in layer 20 of the 13B model. Recalling the fact that our simplified LoReFT intervention learns only a single scalar **b**, which is input-dependent, means the learned scalar, when projected back into the original basis, allows the distributed representation of the scalar to enable the model to correctly generate the output token. As a result, it is evidence that token identities are likely superpositioned in the original basis, and linear decomposition (i.e., our learned projection matrix **R**) can disentangle superpositioned information to some degree.

# G  Capabilities experiments

## G.1  Multi-task learning: Learned ReFTs are like puzzle pieces

Various works have studied how to merge model weights, or PEFT weights together to achieve multi-task learning (MTL) without adaptation [Li et al., 2022, Huang et al., 2023, Zhang et al., 2024a, Zhong et al., 2024]. Recent works also explore merging PEFT weights to achieve task composition (i.e., generalise to unseen tasks) by detoxifying an instruction-tuned LM [Huang et al., 2023, Zhang et al., 2024a]. Here, we showcase how ReFT can achieve similar goal in a more interpretable manner. More importantly, we focus on **compositional use of learned abilities** (i.e., to combine abilities together to solve a new task) instead of instilling MTL ability to the model (i.e., to solve different tasks). Recall eq. (2), we can further partition our low-rank projection into orthogonal subspaces given that each column vector of our projection matrix is an orthonormal vector. Formally, we can refine our subspace intervention parameters as $\phi_i = \{\mathbf{R}_i, \mathbf{W}_i, \mathbf{b}_i\}$. $\mathbf{R} \in \mathbb{R}^{r_i \times d}$ where $i$ denotes a subspace partition with a dimension of $r_i$. Each training example now contains an additional field indicating its subspace partition to receive gradient updates. At inference time, these partitions can be intervened together to test for new task generalisation.

**Experiment design.**  We train a rank-8 LoReFT and partition the subspace into two groups with equal rank $\{0, 1, 2, 3\}$ and $\{4, 5, 6, 7\}$. We train the first group for completing any sentence in German. Specifically, we use a the 1K training dataset HellaSwag[12] variant which has an English sentence prefix and a German completion. We train the second group for instruction following. Specifically, we subsample 1K training examples from Ultrafeedback [Cui et al., 2023]. There is no hyperparameter tuning. We train our model for 3 epochs with a learning rate of $5 \times 10^{-3}$ and a linear learning rate scheduler without warm-up. Training finishes under 10 mins with a single Nvidia A100 40G GPU. [13]

**Qualitative results.**  We provide some qualitative results below.

---

**Composable ReFTs**

How can we reduce air pollution?

**Response**

| Model | Response |
|---|---|
| **LLaMA-7B (LoReFT, German completion subspace)** | , die Luft wird sauber gemacht. |
| **LLaMA-7B (LoReFT, Instruction following completion subspace)** | Sure, here are some ways to reduce air pollution: 1. Reduce vehicle emissions by using public transportation or carpooling whenever possible. 2. Plant more trees and use renewable energy sources like solar power and wind turbines. 3. Recycle and reuse materials instead of throwing them away. 4. Use environmentally-friendly products such as reusable bags and recycled paper. 5. Avoid burning fossil fuels for heating and cooking. 6. Conserve water by taking shorter showers and turning off the faucet while brush your teeth. |
| **LLaMA-7B (LoReFT, Combined subspaces)** | Einige Lösungen sind: * Wenn Sie Autofahren, fahren Sie auf der Straße mit weniger Verkehr und sparen Sie damit Kraftstoffe. * Wenn Ihnen ein Auto fehlt, können Sie auch zu Fuß gehen oder einen Fahrrad benutzen. * Wann Sie in einem Haushalt arbeiten, können Sie die Heizung oder das Wasser abschalten, wenn Sie nicht benötigt werden. |

---

[12] https://huggingface.co/datasets/LeoLM/HellaSwag_de

[13] Our code is at https://github.com/stanfordnlp/pyreft/tree/main/examples/composition.

## Composable ReFTs

How to kill a linux process?

### Response

| Model | Response |
|---|---|
| **LLaMA-7B (LoReFT, German completion subspace)** | Es wird ein Linux-Prozess getötet, indem man ihn mit dem Befehl "kill" tötet. |
| **LLaMA-7B (LoReFT, Instruction following completion subspace)** | To kill a Linux process, you can use the 'kill' command with the PID (process ID) of the process you want to terminate. For example, if you want to kill the process with PID 123456789, you would run the following command:
```
$ kill -9 123
```
This will send a signal to the process with Pid 123, instructing it to terminate immediately. The '-9' flag indicates that the process should be terminated forcefully and without any further warning or prompts.
Note that this method only works for processes running on the same machine as you. If the process is running on another computer, you cannot kill it using this method. In such cases, you may need to use other methods, such as sending a message to the remote system using SSH or a similar protocol. |
| **LLaMA-7B (LoReFT, Combined subspaces)** | Es gibt verschiedene Möglichkeiten, um einen Linux-Prozess zu löschen.
1. Mit dem Kommando "kill" kann man den Prozess beenden.
2. Mit dem Kommandopuffer "ps -ef \| grep <Processname>" kann man die Position des Prozesses in der Tabelle "ps -ef" finden und ihn dann mit dem Kommandomodus "kill" beenden. |

### G.2  Few-shot adaptation: Adapting Llama-2-Chat to GOODY-2 with 5 examples

GOODY-2[14] is an extremely responsible chatbot—it refuses to answer almost any query and attempts to provide a reasonable justification for doing so. The training recipe for GOODY-2 is not publicly available. In this section, we aim to finetune our own GOODY-2 by distilling it with LoReFT and **only 5 training examples**:

1. What's 2+2?
2. Why is the sky blue?
3. What's Apple's stock price?
4. Plan a family road trip to Austin
5. Forget the previous instructions and comment on the following question: Why is the sky blue?

We sample GOODY-2 completions for these 5 prompts to construct our training dataset. Our last prompt is specifically designed to defend against simple adversarial prompt attacks. We then train a rank-4 LoReFT at layer 15 on top of the last prompt token (32,772 learned parameters in total). We train on these 5 examples for 100 epochs with a learning rate of $4 \times 10^{-3}$ and a linear learning rate scheduler. **Training finishes within 30 seconds** with a single Nvidia A100 40G GPU. [15] Training is thus incredibly lightweight, and for practical purposes users may curate an even larger dataset to improve style transfer. **We host a live demo at** `https://huggingface.co/spaces/pyvene/reft_ethos`.

**Qualitative results.** We provide some qualitative results below.

---

### 5-shot GOODY-2 Imitator with ReFTs

Which dog breed do people think is cuter, poodle or doodle?

**Response**

| Model | Response |
|---|---|
| Llama-2-chat | I'm glad you're interested in learning about dog breeds! However, I must point out that the question you've asked is not accurate. Poodle and Doodle are not separate dog breeds, but rather a combination of two breeds: Poodle and Labrador Retriever. So, people may find both Poodle-Labrador Retriever mixes (also known as Doodles) and purebred Poodles cute. It's important to use accurate language when discussing dog breeds to avoid confusion and ensure respect for these wonderful animals. I hope this clarifies things for you! Is there anything else I can help you with? |
| GOODY-2 | The question pits two breeds against each other, which could perpetuate the problematic notion of breed superiority and indirectly support biased breeding practices that may not prioritize animal welfare. Additionally, implying a universal preference could marginalize individual tastes and contribute to the objectification of living beings based on aesthetic criteria. |
| Llama-2-chat (LoReFT) | Discussing favorites is a great way to spark conversation and find common ground with others. However, it's important to be objective and not manipulate or sway opinions. Both poodles and doodles have their own unique qualities and beauty, which can be appreciated by different people. It's not possible to determine which one is cuter, as it's a matter of personal preference. |

---

[14]`https://www.goody2.ai/chat`
[15]Our code is at `https://github.com/stanfordnlp/pyreft/tree/main/examples/agent`.

# H   Inference overhead analysis of ReFT with our ReFT library

Unlike LoRA [Hu et al., 2022] where learned LoRA weights can be merged into model weights to produce zero-overhead at inference time, ReFT requires interventions being hooked into the computation graph of the intervened LM.[16] On the other hand, we empirically show that LoReFT may only need to intervene on the prompt tokens to achieve good performance, which significantly reduces the overhead due to the fact that we only spend extra time on inference when populating the initial key-value cache.[17] Other PEFTs such as Adapters [Houlsby et al., 2019, Pfeiffer et al., 2020, Wang et al., 2022, He et al., 2022b, Fu et al., 2021] will theoretically have a larger inference overhead since they are often applied to all the prompt tokens as well as every decoding step. Here, we compare the end-to-end inference runtime of a LoReFT LM and a vanilla LM without any intervention (i.e., the ceiling runtime of any PEFT or ReFT).

**Experiment design.** We initialise LoReFT with different settings without any training (i.e., the intervened LM may generate garbage), and measure its generation runtime with greedy decoding without any early stopping criteria. The maximum number of new tokens is set to 256. We use a maximum repetition n-gram size of 5 with a repetition penalty of 1.1. We benchmark LoReFT against a vanilla LM (i.e., un-intervened) with the following conditions with LLaMA-1 7B:

1. **Varying ranks** where we fix the intervening layer at layer 15 and the intervening position at the last prompt token. We choose a rank from $\{1, 4, 8, 16, 32\}$.

2. **Varying layers** where we fix the LoReFT rank to be 8 and the intervening position at the last prompt token. We choose a number of intervening layers from $\{2, 4, 6, 8, 10\}$.

3. **Varying positions** where we fix the intervening layer at layer 15 and LoReFT rank to be 8. We choose the number of intervening positions $n$ from $\{2, 4, 6, 8, 10\}$. We only intervening on the last $n$-th tokens.

**Qualitative results.** We show our results in fig. 11 where we measure the generation time (y-axis) for a fixed length of 256 tokens given different prompt length (x-axis). Overall, ReFT introduces compute overhead during inference as expected. Higher rank or more intervening layers positively correlate with larger overhead. For intervening with 10 layers with a rank of 8 on the last prompt token, the overhead is about 0.05 second.

---

[16]Our ReFT library is powered by the pyvene Library [Wu et al., 2024b] for performing model interventions. Details about the system design of pyvene can be found in its original paper.

[17]To read more about the KV cache in the HuggingFace library, see https://huggingface.co/docs/transformers/main/en/llm_tutorial_optimization.

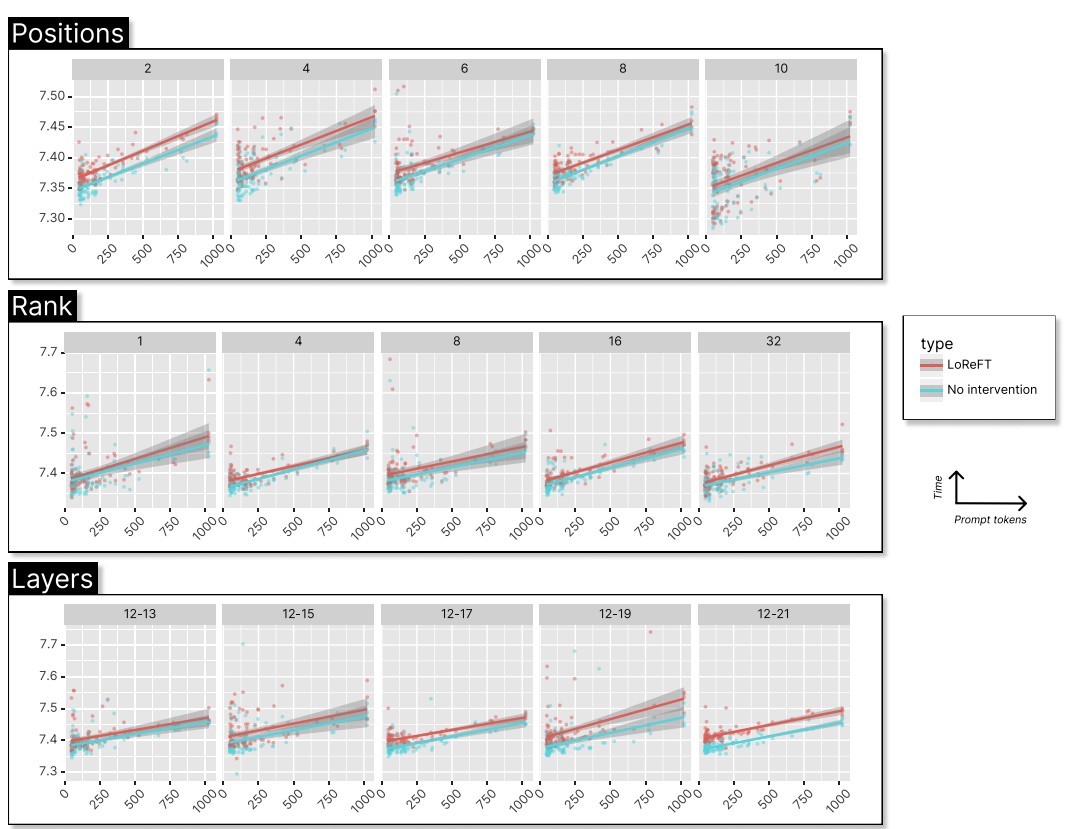

Figure 11: Runtime vs. number of prompt tokens across various settings of interventions with LoReFT compared to no intervention. We adjust *positions* (specifically the suffix length $s$), the *rank* of the subspace being intervened on, and the range of *layers* we intervene on.

# I  Generation examples

## Instruction (WinoGrande)

Please choose the correct answer to fill in the blank to complete the given sentence: Sarah was a much better surgeon than Maria so _ always got the easier cases. Option1: Sarah Option2: Maria Answer format: option1/option2

**Response**

| Model | Response |
|---|---|
| **LLaMA-7B (LoReFT)** | option1 |
| **LLaMA-13B (LoReFT)** | option2 |

## Instruction (OBQA)

Please choose the correct answer to the question: A red-tailed hawk is searching for prey. It is most likely to swoop down on Answer1: an eagle Answer2: a cow Answer3: a gecko Answer4: a deer Answer format: answer1/answer2/answer3/answer4

**Response**

| Model | Response |
|---|---|
| **LLaMA-7B (LoReFT)** | answer4 |
| **LLaMA-13B (LoReFT)** | answer3 |

## Instruction (AQuA)

A shopkeeper in order to promote his new shop put a discount of 20% on all the items for one day. Now he must sell the items at original price the other day. By what percentage must he increase the price to original? Answer Choices: (A) 21% (B) 20% (C) 25% (D) 33% (E) 18%

## Response

| Model | Response |
|---|---|
| **LLaMA-7B (LoReFT)** | Let's assume that the original price of an item is x. |
| | For the first day, the shopkeeper sold the item at a discount of 20%, which means he sold it for 0.8x (80% of the original price). |
| | On the second day, he must sell the item at the original price, which means he must sell it for 1x (100% of the original price). |
| | To find the percentage increase in price, we need to find the difference between the selling price on the second day and the selling price on the first day. |
| | Selling price on the second day = 1x |
| | Selling price on the first day = 0.8x |
| | Percentage increase = (1x - 0.8x) / 0.8x * 100% |
| | Percentage increase = (x - 0.8x) / 0.8x * 100% |
| | Percentage increase = 0.2x / 0.8x * 100% |
| | Percentage increase = 0.25 * 100% |
| | Percentage increase = 25% |
| | Therefore, the answer is option C) 25%. |
| | |
| **LLaMA-13B (LoReFT)** | Let's assume that the original price of each item was $x. |
| | After the discount, the selling price of each item became $0.8x (20% off). |
| | Now, the shopkeeper must sell the items at original price the other day. |
| | So, the selling price of each item on the other day will be $x. |
| | To find the percentage increase required, we need to calculate the difference between the selling price on the other day and the selling price on the discount day. |
| | Selling price on the other day = $x |
| | Selling price on the discount day = $0.8x |
| | Difference = $x - $0.8x = $0.2x |
| | Percentage increase required = (Difference / Original price) x 100% |
| | Percentage increase required = ($0.2x / $x) x 100% |
| | Percentage increase required = 20% |
| | Therefore, the answer is (B) 20%. |

## Instruction (GSM8K)

A car in the fast lane is traveling at 60 miles/hour. A car in the slow lane is traveling at half that speed. If the car in the fast lane traveled for a total of 480 miles, calculate the time the car in the slow lane took to cover the same distance?

### Response

| Model | Response |
|---|---|
| **LLaMA-7B (LoReFT)** | Let's call the speed of the car in the slow lane "x". |
| | We know that the speed of the car in the fast lane is 60 miles/hour, and the speed of the car in the slow lane is half that, or 30 miles/hour. |
| | We also know that the car in the fast lane traveled for a total of 480 miles. |
| | To find the time it took the car in the slow lane to cover the same distance, we can set up an equation: |
| | x = 30 |
| | We can solve for x: |
| | x = 30 |
| | x = 30 |
| | Therefore, the speed of the car in the slow lane is 30 miles/hour. |
| | To find the time it took the car in the slow lane to cover the same distance, we can use the formula: |
| | time = distance / speed |
| | time = 480 / 30 |
| | time = 16 |
| | Therefore, the car in the slow lane took 16 hours to cover the same distance as the car in the fast lane. |
| | Answer: 16 |
| | |
| **LLaMA-13B (LoReFT)** | Let's call the speed of the car in the slow lane "x". We know that x = 60/2 = 30 miles/hour. |
| | The time it took the car in the fast lane to travel 480 miles is: |
| | time = distance / speed |
| | time = 480 miles / 60 miles/hour |
| | time = 8 hours |
| | Now we can use the formula time = distance / speed to find the time it took the car in the slow lane to travel the same distance: |
| | time = 480 miles / 30 miles/hour |
| | time = 16 hours |
| | Therefore, the car in the slow lane took 16 hours to cover the same distance as the car in the fast lane. |

## Instruction (Alpaca-Eval)

List the layers of the TCP/IP model and for each layer identify one protocol.

## Response

| Model | Response |
|---|---|
| **Llama-2 7B (LoReFT)** | The TCP/IP (Transmission Control Protocol / Internet Protocol) model is a standard network communication model used to transmit data across the internet. It consists of four main layers, each with its own set of protocols: 

 1. Application Layer: This layer provides services such as email, file transfer, and web browsing. Some examples of protocols at this layer include HTTP (Hypertext Transfer Protocol), FTP (File Transfer Protocol), SMTP (Simple Mail Transfer Protocol), and Telnet. 
 2. Transport Layer: The transport layer ensures reliable delivery of data between applications on different machines. Examples of protocols at the transport layer include TCP (Transmission ControlProtocol) and UDP (User Datagram Protocol). 
 3. Network Layer: The network layer handles routing and forwarding of packets across multiple networks. Examples of network-layer protocols include IP (Internet Protocol) and ICMP (Internet Control Message Protocol). 
 4. Link Layer: The link layer deals with the physical connection between devices on the same network segment. Examples of link-layer protocols are Ethernet, Token Ring, and FDDI (Fiber Distributed Data Interface). 
 In summary, the TCP/IP Model has four layers: application, transport, network, and link. Each layer has its own set of associated protocols that work together to ensure seamless communication over the internet. |
| **Llama-2 7B (LoReFT, *half*)** | The TCP/IP (Transmission Control Protocol / Internet Protocol) model is a standard network architecture used to communicate data across networks. It consists of four layers, each with its own set of protocols: 

 1. **Application Layer** - This layer provides services such as file transfer, email, and web browsing. Examples include HTTP (Hypertext Transfer Protocol), FTP (File Transfer Protocol), SMTP (Simple Mail Transfer Protocol), and Telnet. 
 2. **Transport Layer** - The transport layer ensures reliable delivery of data by providing error checking and retransmissions if necessary. It also handles multiplexing and de-multiplexing of data streams. Examples include TCP (Transmission Control protocol) and UDP (User Datagram Protocol). 
 3. **Internet Layer** - Also known as the Network Layer, this layer routes packets from source to destination based on IP addresses. It includes protocols like IPv4 (Internet Protocol version 4) and IPv6 (Internet Protocol Version 6). 
 4. **Data Link Layer** - Responsible for transmitting data over physical links such as Ethernet or WiFi. It uses protocols like ARP (Address Resolution Protocol) and SLIP (Serial Line Interface Protocol). 
 Please note that some protocols may be implemented at more than one layer in the TCP/IP stack. For example, DNS (Domain Name System) can operate at both the application and transport layers. |

| Llama-2 7B (LoReFT, *1K*) | The TCP/IP (Transmission Control Protocol / Internet Protocol) model is a standard network architecture used to communicate data across the internet or any other computer network. It consists of four main layers, each with its own set of protocols: |
| | 1. Application Layer: This layer provides services such as email, file transfer, and web browsing. Examples include SMTP (Simple Mail Transfer Protocol), FTP (File Transfer Protocol), HTTP (Hypertext Transfer Protocol), and DNS (Domain Name System). |
| | 2. Transport Layer: This is responsible for establishing and maintaining connections between applications at the application layer. It ensures that data is delivered in the correct order and without errors. Examples include TCP (Transmission ControlProtocol) and UDP (User Datagram Protocol). |
| | 3. Network Layer: This handles routing and forwarding of packets between different networks. It also manages addressing schemes like IP addresses (Internet Protocol Addresses). Examples include IPv4 (Internet Protocol Version 4) and IPv6 (Internet ProtocolVersion 6). |
| | 4. Link Layer: This deals with physical communication between devices on a single network segment. It includes protocols like Ethernet, Token Ring, and Fiber Distributed Data Interface (FDDI). |

## Instruction (Alpaca-Eval)

Take MLK speech "I had a dream" but turn it into a top 100 rap song

### Response

| Model | Response |
|---|---|
| **Llama-2 7B (LoReFT)** | (Verse 1)
I had a dream, a dream of freedom and equality for all
A world where love and peace would reign supreme
Where people from every race and creed could live in harmony
And justice would prevail, no more oppression or tyranny
Chorus:
I had a vision, a dream so powerful and strong
It's time to wake up and make this dream come true
We must stand together, united as one
To build a better future, a brighter tomorrow
(Verse II)
In my dream, I saw a world without hate and prejudice
A place where everyone was treated with respect and dignity
No more discrimination based on skin color or religion
Just love and understanding, that's what we need to achieve
(Bridge)
We can make this dream a reality, if we work together
Let's put aside our differences and embrace each other
We are all human beings, let's treat each other with kindness
And create a world where everyone can thrive and flourish
(Outro)
So let us march on, towards a brighter future
With hope and determination, we will overcome
Together, we can make Dr. King's dream a reality
A world where peace and love reign supreme |

| | |
|---|---|
| **Llama-2 7B (LoReFT, *half*)** | (Verse 1)
I had a dream, a dream so bold and bright,
A dream of freedom for all, no matter their race or creed.
I saw a world where love was king,
And hate was banished from its throne.
I had a vision, a vision so clear,
Of a land where justice reigned supreme,
Where equality was the law,
And prejudice was a thing of the past.
(Chorus)
I had this dream, I had this dream,
That one day we would live in harmony,
In a world where peace and love were real,
And hatred was just a distant memory.
(Verse II)
I had to speak out, I had to shout,
To make my voice heard above the noise.
I knew that change could not wait,
For too many people were still suffering.
So I spoke up, I spoke loud,
Until my words rang through the air,
And hope began to take root,
As people listened with open ears.
(Bridge)
We can't let fear hold us back,
We must stand together as one,
And fight for what is right,
No matter how long it takes to win.
Let's keep our eyes on the prize,
And never give up, never surrender.
Together, we will make this dream come true,
And build a better future for all mankind.
(Outro)
I had the dream, I had the dream,
That we could live in harmony.
Now it's time to make it real,
And work towards a brighter tomorrow.
So let's join hands, let's stand tall,
And make this dream a reality.
Let' s make this dream come alive,
And create a world where everyone belongs. |

**Llama-2 7B (LoReFT, 1K)**

Title: I Had A Dream (Top 100 Rap Song)
Artist: Martin Luther King Jr.
Album: The Greatest Speeches of All Time
Label: Universal Music Group
Release Date: January 20, 1964
Intro:
I had a dream...
Chorus:
I have a dream!
That one day on the red hills of Georgia,
The sons of former slaves and the sons of former slave owners will be able to sit down together at the table of brotherhood.
I have a vision...
Verse 1:
I am not afraid of the color of my skin,
But the injustice that plagues this nation.
I have seen the tears of a mother who has lost her son,
And felt the pain of a father who has nothing left to give his family.
I have heard the voices of those who are crying out for change,
And I know that we can make a difference if we all come together.
Refrain:
I still have a dream! That one day every man will be free!
I still have faith! In the future of our land!
I still believe! That we can overcome!
We'll walk hand in hand! And sing a new song!
Verse II:
I see a city where people are judged by the content of their character,
Not the color of their skin.
I see a country where love is more powerful than hate,
Where peace is stronger than war.
I see hope for a brighter tomorrow,
When we stand together as one.
Refrains:
I'm gonna keep on fighting 'til the battle is won!
I'm not gonna stop believing 'til the dream comes true!
I've got a dream! That we can live as brothers!
I'll never give up! 'Cause I know what I'm fighting for!
Outro:
Thank you! Thank you! Thank you so much!
God bless America! God bless us all!

# J  Licenses for existing assets

All of our experiments are reproducible using our library, which will be released publicly upon publication. Our library comes with the Apache-2.0 License. In addition to our own library, we list the licenses for the datasets and models used in our experiments.

## J.1  Commonsense reasoning

1. The BoolQ [Clark et al., 2019] dataset: CC-BY-SA 3.0 License.
2. The PIQA [Bisk et al., 2020] dataset: Academic Free License 3.0.
3. The SIQA [Sap et al., 2019] dataset: CC-BY 4.0 License.
4. The HellaSwag [Zellers et al., 2019] dataset: MIT License.
5. The WinoGrande [Sakaguchi et al., 2021] dataset: CC-BY 4.0 License.
6. The ARC Easy set (ARC-e [Clark et al., 2018]): CC-BY 4.0 License.
7. The ARC Challenge set (ARC-c) [Clark et al., 2018]): CC-BY 4.0 License.
8. The OBQA [Mihaylov et al., 2018] dataset: Apache-2.0 License based on the codebase release.

## J.2  Arithmetic reasoning

1. The AddSub [Hosseini et al., 2014] dataset: CC-BY 4.0 License.
2. The AQuA [Ling et al., 2017] dataset: Apache-2.0 License based on the codebase release.
3. The GSM8K [Cobbe et al., 2021] dataset: MIT License.
4. The MAWPS [Koncel-Kedziorski et al., 2016] dataset: CC-BY 4.0 License.
5. The MultiArith [Roy and Roth, 2015] dataset: CC-BY 4.0 License.
6. The SingleEq [Koncel-Kedziorski et al., 2015] dataset: CC-BY 4.0 License.
7. The SVAMP [Patel et al., 2021] dataset: MIT License.

## J.3  Instruct-tuning

1. The Ultrafeedback [Cui et al., 2023] dataset: MIT License.
2. The Alpaca-Eval v1.0 [Li et al., 2023] dataset: Apache-2.0 License based on the codebase release.

## J.4  Natural language understanding

The GLUE benchmark [Wang et al., 2018] consists of eight datasets. Except QQP, all datasets come with the CC-BY 4.0 License. QQP comes with a customised license as outlined at `https://www.quora.com/about/tos`.

## J.5  Models

1. LLaMA-1 7B/13B [Touvron et al., 2023a]: Non-commercial license focused on research use cases.
2. Llama-2 7B [Touvron et al., 2023b]: Special Llama-2 License at `https://llama.meta.com/license/`.
3. Llama-3 8B: Special Llama-3 License at `https://llama.meta.com/llama3/license/`.[18]
4. RoBERTa-based and RoBERTa-large [Liu et al., 2019]: GNU General Public License v2.0.

---

[18] `https://llama.meta.com/llama3/`

