# OpenReview forum: "ReFT: Representation Finetuning for Language Models"
_NeurIPS.cc/2024/Conference — NeurIPS 2024 spotlight_

### Official Review · Reviewer_333n · 2024-07-09

**Soundness:** 3
**Presentation:** 3
**Contribution:** 4
**Rating:** 7
**Confidence:** 3

**Summary:**

The paper introduces Representation Finetuning (ReFT) - a family of methods to learn interventions directly on model representations, rather than model weights. The authors compare ReFT to Parameter Efficient Finetuning (PEFT), and find that it yields similar performance while being significantly more parameter-efficient.

**Strengths:**

- The paper is well-written.
	- The main ideas in the paper are clear and easy to understand.
- Novel and impactful contribution.
	- The paper presents a new paradigm of learning modifications to representations, rather than learning modifications to weights. This new paradigm is more parameter-efficient, while appearing to have a similar level of expressivity. PEFT methods have been incredibly impactful, and I could see ReFT being similarly impactful.

**Weaknesses:**

- Lack of representation-editing-based baselines
	- The paper focuses on comparing ReFT to PEFT methods. Appendix B discusses existing representation-editing methods, and casts them in terms of the ReFT framework.
	- I think it would strengthen the paper to compare ReFT and DiReFT to these existing representation-editing methods. I am curious to understand the difference in performance of an intervention learned by gradient descent (e.g. ReFT) vs an intervention learned by contrastive pairs (e.g. activation addition, RepE).
- Some unclear presentation
	- Line 102
		- What does "the hidden representation created at row $i$ and column $k$" refer to? As far as I can tell, this does not integrate with the previous notation defined in the second paragraph of Section 3. I assume this refers to layers and positions - if this is the case, then it would be clearer to say so explicitly.
	- Inconsistent variable names
		- I think the presentation would be clearer if variable names were used across sections.
		- Section 3.1 uses $\mathbf{b}$ to represent a hidden state, whereas Section 3.2 uses $\mathbf{b}$ to refer to a bias vector. I think this can be easily fixed by using $\mathbf{h}$ to represent the hidden state in Section 3.1.
		- Section 3 uses $m$ to represent the number of layers, whereas Section 3.2 uses $m$ to refer to the length of the output sequence $\mathbf{y}$.
	- Typo in Table 17 (?)
		- LLaMA-7B/DiReFT/AQuA: 221.3.
		- It might be worth double checking your numbers more generally if tables are not generated by code.
- Other suggestions
	- Section 3.2
		- The paper could benefit from a motivation of the LoReFT expression, and how it was selected over the other expressions mentioned in Appendix E.
	- Define dropout more explicitly
		- What does dropout refer to in this case? My assumption is that with probability $p$, the intervention is not performed. But in the cases where the intervention is not performed ("dropped out"), what is being optimized?
	- Include baseline generations in Appendix I
		- Only LoReFT outputs are given in Appendix I - it's hard to interpret these generations without having baselines to compare them to. For example, even examining the difference between the baseline (no intervention) and LoReFT would be helpful, particularly for the long-form generations.

**Questions:**

- Practical recommendations for using ReFT:
	- Appendix E mentions that there is not one clear best expression of ReFT. Which variation would the authors recommend individuals use, and why?
	- Is there a recommended methodology for determining hyperparameters in practice?
	- In what scenarios should one use PEFT vs ReFT vs other inference-time interventions?

**Limitations:**

- The authors address the following limitations:
	- The classes of models evaluated is limited.
	- Hyperparameter selection seems fairly complicated, and automating this selection will be valuable for future adoption of the methodology.

---

> ### Author Rebuttal · Authors · 2024-08-04
>
> Thanks so much for raising these great questions and providing helpful feedback!
>
> ### RepE baseline.
>
> We agree that gradient-free methods such as activation addition or RepE could be effective in steering models for tasks such as style transfer [4]. On the other hand, **we argue that it could be hard for these methods to steer models to achieve general finetuning goals** (e.g., steer a model to become a binary sentiment classifier or steer a model to answer multiple choice questions). These methods usually rely on statistical methods (e.g., PCA) to propose a set of static steering vectors based on a few-shot training dataset (i.e., usually with a handful of training examples). Mathematically, **these methods have the same expressivity as BitFit** [5] (only learning the bias term of LMs) yet BitFit is learned with SGD on large datasets. Meanwhile BitFit, although with much lower trainable parameter count (comparable to ReFT), usually underperforms compared to other PEFTs (see Table 4 on pg. 9). However, we agree that learning-free methods such as RepE have other unique advantages:
>
> - it could be very effective when the steering objective is more generic (e.g., style transfer for a chat-model) without heavy shifts in model’s behavior;
>
> - it could be very useful for LM personalization applications with very limited training source and do not require high precision.
>
> ### Need to include the baseline generations.
>
> Thanks for raising this issue. We agree. For all examples in Appendix I, we will add in the original mode outputs. **We include one example from GSM8K in our attached rebuttal pdf**.
>
> ### The best expression of ReFT.
>
> Indeed, there is no clear winner for math reasoning benchmarks. Here are some additional pointers in terms of choosing different intervention functions:
>
> - **DiReFT removes the orthogonality constraint which improves the memory footprint** and trades compute efficiency (i.e., DiReFT trains faster) for a slight drop in performance.
>
> - **LoReFT generally converges quicker** and is less sensitive to learning rate and other hyperparameters partly due to its orthogonality constraint.
>
> - **The orthogonality constraint offers composability and interpretability**. Given the constraint, subspaces are orthogonal to each other (i.e., changing in one subspace should not affect others). As a result, we think this gives a nice property for us to compose LoReFT together at inference-time. We explore this a bit in Appendix G.1..
>
>
> ### Hyperparameter selection with ReFT.
>
> **We address this question in our general responses** by providing additional details on the hyperparameter searching process of ReFT!
>
>
> ### PEFT vs ReFT vs other inference-time interventions.
>
> Thanks for the question! It would be best for users to benchmark these methods for a specific domain. If we were allowed to guess, here are some pointers:
>
> - Inference-time intervention or activation addition does not require training. If the use case is not mission critical, one could use these methods for quick turnaround and showcase.
>
> - In general, since ReFT allows gradient descent, it should be more effective than non-training methods since it actively searches causal pathways to steer the model. ReFT also works with quick adaptation (e.g., n-shot training where n <= 10) as shown in G.2..
>
> - The composability of ReFT could be better (e.g., combining a set of directional steering together such as changing the tone, Language and length of the generating text.) compared with PEFT and ITI. We include a preliminary exploration of composability in Appendix G.2..
>
> Again, we feel like this is an open-ended question. And introducing ReFT definitely pushes the community to think about the differences among these methods.
>
> ### The classes of models evaluated is limited.
>
> **We addressed this question in our general responses** by applying ReFT to other models and tasks!
>
>
> ### Unclear presentation.
>
> Thanks for all these suggestions! We will address the following items in camera-ready.
>
> > Line 102: What does "the hidden representation created at row and column" refer to?
>
> Sorry about the confusion here. Row and column map to the layer and position of the intervening residual stream. We will remove these two redundant notations $i$ and $k$, and rewrite the current sentence as: *“Let b be the hidden representation created at a specific residual stream located at a specific position and layer., …”*
>
> > Variable names were used across sections and typos.
>
> We will make the following changes:
>
> - We will revisit our notations in Sec. 3 and keep a consistent format in our next revision. Please feel free to raise additional suggestions, and we will incorporate them.
>
> - We will replace our notation for $b$ in Eqn. 1 and related parts with $h_b$ to represent the base representation.
>
> - We will replace $m$ in Sec. 3.2 with $k$ to avoid overloading our symbols.
>
> - We also noticed this typo ("LLaMA-7B/DiReFT/AQuA: 221.3") after we submitted our draft. The correct entry should be 21.3. We will update the number, and check existing ones in our next revision. Our result tables (i.e., not the hyperparameter tables) are semi-automatically generated, but human error is still possible)
>
>
> > Other suggestions on writings.
>
> We will make the following changes:
>
> - We will motivate how we come up with the current LoReFT formula better. We did cut a bit of text on this from our earlier draft due to length concerns. We will motivate LoReFT better from the interpretability works by bridging Sec 3.2 and Sec 3.2 together better in the next revision.
>
> - We will clarify the use of dropout in Eqn. (2) and (3).
>
> [4] Zou et. al., 2023, “Representation Engineering: A Top-Down Approach to AI Transparency”
>
> [5] Zaken et. al., 2022, “BitFit: Simple Parameter-efficient Fine-tuning for Transformer-based Masked Language-models”

---

> > ### Comment · Reviewer_333n · 2024-08-08
> >
> > I've read the author rebuttal and the general response. I thank the authors for their diligent engagement. I think the proposed writing edits will improve the manuscript. I elect to maintain my overall score.

---

### Official Review · Reviewer_ddW1 · 2024-07-12

**Soundness:** 4
**Presentation:** 4
**Contribution:** 4
**Rating:** 9
**Confidence:** 4

**Summary:**

This work proposes a novel method for fine-tuning language models (LM) called Representation Fine-tuning (ReFT), which updates only a small number of parameters. Unlike existing parameter-efficient fine-tuning methods such as LoRA, ReFT enables fine-tuning with minimal parameter updates by learning small interventions in the representation of specific layers and token positions. The authors introduce Low-rank Linear Subspace ReFT as a method within ReFT, demonstrating experimentally that it can fine-tune LMs effectively with fewer parameter updates compared to existing parameter-efficient fine-tuning methods.

**Strengths:**

- **Originality**: While representation engineering has been utilized in various works, applying it to parameter-efficient fine-tuning is unprecedented. The motivation from interpretability works is also interesting. From the perspective of originality, this work is commendable.
- **Quality**: The proposed method’s effectiveness is empirically validated on multiple LMs based on Llama and RoBERTa across approximately 20 datasets and four tasks, proving its practical applicability. Additionally, the authors provide extensive experimental results with various hyperparameters in the appendix, offering sufficient reference data for future users and making it easy to identify scenarios requiring caution. Thus, this paper can be considered a complete piece of work.
- **Clarity**: The paper is written very clearly, including appropriate figures to facilitate easy implementation by the reader.
- **Significance**: This paper holds significant value as it suggests a more effective way to fine-tune LMs through interventions in representation rather than model weights, unlike existing PEFT methods. It can serve as a drop-in replacement for the widely used LoRA, potentially having a substantial impact on future LM fine-tuning research.

**Weaknesses:**

- **Quality**: Although the extensive experimental results in numerous settings sufficiently demonstrate the method's significance compared to existing methods, including results from models other than Llama, such as Mistral or Phi, would emphasize the method's applicability in various scenarios.
- **Significance**: The need for extensive hyperparameter optimization to decide which layer and position of hidden representation to apply the intervention function is a potential weakness. This issue is well explained and mentioned in the Limitations section.

**Questions:**

### Questions

Practically speaking, when do the authors believe this method should be attempted instead of LoRA? In other words, in what situations is the LoREFT most appropriate? Based on the experimental results, it appears that using the LoREFT might be unsuitable for achieving high performance on inference tasks such as GSM8K.

**Limitations:**

The authors adequately addressed the limitations in Section 5.

---

> ### Author Rebuttal · Authors · 2024-08-04
>
> Thanks for assessing our paper to be a significant contribution, and for your question!
>
> ### ReFT with LMs other than LLaMAs.
>
> **We addressed this question in our general responses** by exploring other model types such as Mistral and Phi! As shown by our initial results, both LoReFT and DiReFT work for other types of LMs.
>
> ### Hyperparameter selection process of ReFT.
>
> **We took up this suggestion in our general responses** by providing more details on the current hyperparameter searching process.
>
> ### Practically, when to choose ReFT over LoRA?
>
> Thanks for raising the question! We wish to include more of these insights in the main text with additional space allowed in the next revision. For now, here are some practical guide of using ReFT:
>
> - **LoReFT works better when the base LM is strong**: in our experiments, we usually find ReFT scales with the quality of the base LM for harder generation tasks. For instance, the gap between ReFT and LoRA is much smaller when applying ReFT to larger LLaMA models for our math reasoning benchmarks as shown in Table 2.
>
> - **LoReFT (or ReFT) is composable by its nature**: LoReFT localizes task-dependent interventions into orthogonal subspaces: you can partition the subspaces of a single LoReFT intervention for different tasks. Specifically, you can train different subspaces for different tasks, and compose them together to learn a combined skill. We showed some initial results in Appendix G.2.. Although LoRA weights can be squished together, ReFT is much more interpretable. Additionally, the number of intervenable representations are abundant in LLMs. As a result, it becomes much more feasible to overload and stack interventions together in a zero-shot fashion.
>
> - **Practically, ReFT could be a better solution in a multi-tenant finetuned model serving service**: imagine a case where we are serving thousands of finetuned models: for a batch user queries, we want to call different finetuned models. We cannot serve thousands of SFT LMs without enormous costs. One alternative is to have a single base LM and thousands LoRA weights cached in the memory. For this approach, you have to hot-switch between LoRA weights in memory. For ReFT, it potentially becomes much easier since ReFT only needs to intervene on the prompt representations once for the batch, and pass the intervened prompt KV cache to the inference engine without  inference-time overhead. We only realized this after we submit the paper, and we hope to discuss this further in our next revision.
>
> ### To improve ReFT's performance on tasks such as GSM8K.
>
> **We addressed this in our general responses** by improving LoReFT’s performance on math with additional interventions on decoding steps!

---

> > ### Comment · Reviewer_ddW1 · 2024-08-11
> >
> > Thank the authors for their response. I would appreciate it if the details provided in the rebuttal could be included in the final version of the paper. I will maintain my score as I have no further concerns about this work.

---

### Official Review · Reviewer_d959 · 2024-07-12

**Soundness:** 4
**Presentation:** 4
**Contribution:** 3
**Rating:** 6
**Confidence:** 4

**Summary:**

The authors propose an alternative PEFT method based on representation intervention techniques that are used in interpretability research. They evaluate their method in a variety of settings including multiple architectures and finetuning dataset families.

**Strengths:**

* The method presented by the paper uses an order of magnitude fewer parameters than comparable baselines (e.g., LoRA, DoRA) while still maintaining comparable accuracies after finetuning.
* The method achieves consistently stronger performance than the baselines on commonsense reasoning tasks
* Evaluation was thorough: the authors evaluated multiple architectures across scale and multiple dataset families.

**Weaknesses:**

LoRA and other baselines can adjust the number of tunable parameters via the rank parameter. The authors should evaluate specifically how their method compares to the baselines when there are a comparable number of tunable parameters, e.g, by lowering the LoRA rank. The original LoRA paper suggests that performance can sometimes increase (and often, at least, not decrease) when lowering the rank.

Given that the ReFT intervention occurs on a fixed set of positions, the authors should evaluate if ReFT is effective even in long-context settings.

While there are performance improvements on the commonsense reasoning tasks, performance decreases on other tasks (sometimes fairly substantially, e.g. in Table 2), which limits the applicability of this method. While the authors evaluate multiple settings, it would be interesting to present results on a wider class of evaluations and finetuning datasets to evaluate where ReFT would be preferable to other datasets. (Note: While this would be interesting and would make for a stronger paper, I do not believe it is necessary for a solid paper).

**Questions:**

What is the memory footprint of ReFT in comparison with other methods?

Since the intervention occurs on a fixed number of prefix tokens, what happens when the prompt prefix is shorter than the number of intervened positions?

**Limitations:**

The authors adequately address the limitations.

---

> ### Author Rebuttal · Authors · 2024-08-04
>
> Thanks for your constructive feedback and questions!
>
> ### LoRA with fewer parameters has been tried.
>
> We want to clarify that **the baseline numbers we have in our tables are the best performance after hyperparameter tuning** done by the original LLM-Adaptor [3] paper. For instance, this adaptor benchmarking paper searches **the best adaptor location** (out of ranks \{`Attention` only, `MLP` only, `both`\}) as well as best ranks (out of ranks {`4`, `8`, `16`, `32`}) for LoRA. With applying LoRA only to the `Attention` module and rank=`4`, LoRA can get much lower parameter count while sacrificing performance (e.g., see Figure 3 on pg. 5 of their paper). We will clarify this in our next revision.
>
> ### ReFT with long-context tasks.
>
> Thanks for the suggestions! **We took up this suggestion in our general responses** by applying ReFT to a long-context summarization task.
>
>
> ### ReFT with LMs other than LLaMAs.
>
> **We addressed this in our general responses** by applying ReFT to two other LM types: Mistral and Phi-3. We additionally tried to close the gap in the math benchmark by enhancing LoReFT.
>
>
> ### Memory footprint of ReFT.
>
> If we understand the question correctly, **the memory footprint should be largely bounded by the number of training parameters**. Thus, one can use the Params (%) column in our result tables (e.g., Table 1 on pg. 6) to rank the memory footprint for various methods. One thing to note is that LoReFT’s orthogonal constraint does require more memory due to the orthogonalization process compared with DiReFT. However, this might not be the dominating factor.
>
> Moreover, compared against LoRA, **ReFT does require some inference-time overhead** since ReFT is an intervention-based method which cannot merge its learned weights into the original model. Nevertheless, since we constrain ourselves to intervene on the prompt tokens, the overhead is limited (fractional inference time increase is less than 1% in various settings). We provided a detailed analysis in Appendix H on pg. 38, and we hope to highlight this in our next version when more space is allowed in the main text.
>
>
> ### Prompt prefix is shorter than the number of intervention positions.
>
> Thanks for bringing this up, and **this is a great technical question**! We introduced the concept of intervention position padding in our ReFT Python library. In short, it will pad the prompt with a single padding token, and we will perform dummy interventions on this token if needed. The attention mask and loss calculation will bypass this token to make sure other tokens are not affected.
>
> [3] Hu et. al., 2023, "LLM-Adapters: An Adapter Family for Parameter-Efficient Fine-Tuning of Large Language Models"

---

> > ### Comment · Reviewer_d959 · 2024-08-08
> >
> > Thanks for the response and additional experiments.
> >
> > > LoRA with fewer parameters has been tried.
> >
> > Thanks for pointing this out, although I still believe there are some issues with the tuning setup. I think this brings up another issue that I missed in the review: you are using the evaluation from a different paper. How can you be sure that your evaluation setup exactly matches their setup? Further, from their paper, it seems they tune the LoRA rank parameter on math reasoning datasets, which, as your results suggest, have potentially different behavior than other datasets. Also, your method requires tuning which specific layers to which the adapter is applied—what happens if you do this with LoRA? It may be the case that if you apply LoRA to the same set of layers as you use for ReFT may lead to similar performance and a substantial drop in the number of the adapter parameters.
> >
> > In my view, the primary positive aspect of your method is that it requires fewer parameters than LoRA, but generally performs similarly (better on some datasets, worse on others). This is why being extremely careful in your comparison to the number of parameters that LoRA requires is so important.
> >
> >
> > For the other responses—thank you, I appreciate the clarification.

---

> ### Author Response · Authors · 2024-08-08
> **Response from the authors.**
>
> Thanks for raising these methodological points!
>
> Throughout the paper, we report published numbers for other approaches rather than running our own evaluations of those approaches. Our assumption is that the other authors are the most expert in how to get their approaches to work best, and so this provides us with the most ambitious comparison points. We have carefully verified that we are using the same protocols for evaluation and use of the training data.
>
> For the hyperparameter searching comparisons, our method for choosing hyperparameters seems stricter than the norm. In particular, we use GSM8K (also in math reasoning domain) to select our hyperparameters and apply these to both the Math and Commonsense Reasoning benchmarks, to avoid any implicit model selection based on test runs. The other authors use the Math10K train/test split to do model selection.
>
> For the Math and Commonsense Reasoning benchmarks, both LoRA and ReFT are applied to all layers. For each layer, ReFT intervenes on the residual stream (which is weightless and therefore cannot be targeted by LoRA), while LoRA is applied to multiple weight matrices such as the Q/K/V projections. Although the LLM-Adaptor paper did not attempt to tune ranks on the Commonsense Reasoning benchmark, the DoRA paper (a newer variant of LoRA) [6] tried to halve the rank of DoRA (i.e., reducing the parameter count by 50%) and found that performance consistently dropped across all LLaMA models, as we reported in the paper (see DoRA (half) in our Table 1).
>
> We report ReFT evaluations with much smaller parameter counts than the other methods, which would seem to put us at a disadvantage rather than an advantage. We could double-check this by increasing the ReFT parameter count to match the LoRA numbers. We would worry about lowering the LoRA counts and running our own experiments, for the reason we noted above (LoRA advocates might argue for different settings than we would choose).
>
> [6] Liu et. al., 2024, “DoRA: Weight-Decomposed Low-Rank Adaptation”
>
> **Additional clarification:** regarding the evaluation setup, we would like to clarify that we *directly copied* the publicly available codebase for the LLM-Adaptor paper to ensure a fair comparison (e.g., same datasets, evaluation metrics, decoding strategies, etc..).

---

> ### Comment · Reviewer_d959 · 2024-08-12
>
> Thanks for the response!
>
> > Further, from their paper, it seems they tune the LoRA rank parameter on math reasoning datasets, which, as your results suggest, have potentially different behavior than other datasets
>
> To clarify what I mean by this: you find that your methods reasonably underperform e.g., LoRA, on arithmetic reasoning tasks (Table 2), which, as an example, could arise because LoRA has more trainable parameters. If this is true, then when tuning on an arithmetic reasoning task (GSM8K, Math10k, or otherwise), your hyperparameter selection might favor higher LoRA ranks because it adds additional trainable parameters. The most striking claim of your paper is that your method outperforms LoRA specifically on commonsense reasoning tasks with an order of magnitude fewer parameters. It's possible, then, that one could tune LoRA specifically to perform well on these commonsense reasoning tasks by using the same tricks that you use to tune your method (e.g., applying it to specific layers).
>
> I also did read your global response and see that with some additional tweaking you match LoRA on math reasoning datasets, which is interesting—what I'm arguing is simply that it's important to put the same effort into tuning LoRA as your method. For example, as I proposed above, adding an hyperparameter for LoRA (analogous to the one for your method) to specify which layers it is applied to.
>
> With that said I think you have a good paper and I am voting for acceptance, the reason my score isn't higher is because (1) I am not fully convinced that tuning LoRA (e.g., by applying it to specific layers and tuning the rank a bit) could not perform similarly to your method, and (2) because this method might not serve as a good drop-in replacement for LoRA because it underperforms on important datasets (arithmetic reasoning) without extra manual effort.

---

### Official Review · Reviewer_Pcei · 2024-07-14

**Soundness:** 3
**Presentation:** 4
**Contribution:** 3
**Rating:** 6
**Confidence:** 4

**Summary:**

This paper proposes representation finetuning for efficient tuning or intervening for task-specific representations in models while keeping the base model frozen. They define LoReFT and unify several current representation intervention methods under their framework. They conduct extensive experiments on several types of NLP benchmarks and models to demonstrate the efficiency and better understand the effectiveness of ReFT.

**Strengths:**

1. The paper proposes a new and potentially useful paradigm for efficient fine-tuning of model representations for specific tasks. Their proposed LoReFT inherits the merits of previous representation intervention method and operates on a low-rank subspace to control the the representations. They also put LoReFT under a bigger framework of representation intervention and discuss its relationship with other previous method. They provide good insight on understanding the development process of current representation-level interpretability work.

2. Their experiments are extensive and solid. They successfully demonstrated the efficiency of their ReFT, as well as the effectiveness to some extent. I admire that the authors are willing to show the limitation of performance of their method on some types of tasks to faithfully argue the benefits of their method.

3. They have an open-sourced package for reproducing the whole pipeline. They also have detailed documentation of their hyperparameter tuning process. I think this is especially important for new methods like ReFT which I imagine would requires some deep understanding to tune the hyperparameters.

4. They have some interesting intervention examples in the Appendix. They also provide very good ablation of LoReFT design in the Appendix.

**Weaknesses:**

1. I feel the hyperparameter tuning is still pretty heavy althoughtthe authors try to show that they only need to finetune the model on one task from a specific type and use the set of hyperparamters for other tasks in the same type. I expect to see more results on how robust the hyerparameters are across models in the wild. For example, if we find a set of hyperparameters for Llama 7B, will that generalize to Llama2 7B or models of the same size? I also want to see how much the variance is for the best set of hyperparameters for different models and tasks.

2. Sometimes I expect to see how ReFT would fit the current pipeline of SFT + RLHF alignment paradigm. Some results, for example, on the instruction tuning datasets, are lag behind the current SOTA by too far. I'm not saying the method should achieve SOTA but would expect to see the possibility of it to push the new boundaries in the current context of LLM fine-tuning.

**Questions:**

1. What do the prefix and suffix refer here for classification tasks and generation tasks in your hyperparameter tuning descriptions?

2. You might not have experiments on that but just out of curious if ReFT can be used to optimize the objective in preference-based learning like the Bradley-Terry model in DPO and how it would perform.

3. For the ultrafeedback fine-tuning, do you select the best responses to do MLE? As this is usually used as a preference learning dataset.

**Limitations:**

yes

---

> ### Author Rebuttal · Authors · 2024-08-04
>
> Thank you for appreciating our work and raising interesting questions!
>
> ### Generalization of hyperparameters.
>
> Yes, we try to challenge the generalizability of ReFT by testing whether a set of hyperparameters for one task transfers to another as we do hyperparameter search on separate dataset splits. Moreover, we indeed **tried the exact same setting you are proposing here** for our instruction-tuning benchmarks in Sec. 4.4. We select the hyperparameters on LLaMA-1 7B and test those settings with Llama-2 7B without additional hill-climbing (see L248 where we mention this). Although this is not the full picture, our findings do show that selected hyperparameters transfer well across models as well. We will highlight this in our next revision.
>
> ### ReFT with the SFT + RLHF alignment paradigm.
>
> We want to clarify that our focus is not to establish a new SoTA on model alignment compared to SFT+RLHF or DPO. Rather, we seek to offer comprehensive comparisons with other PEFTs.
>
> While ReFT saves parameters and maintains better performance compared with LoRA in our instruction-tuning experiments, **our results do suggest other important application of ReFT** that we wish to discuss in-depth if space is allowed:
>
> - **Much quicker iteration on the alignment pipeline.** For instance, if we want to evaluate the alignment dataset quality of Alapca-15K and ultrafeedback, we could use ReFT to finetune a model and compare the performance instead of SFT. This potentially allows much quicker iterations on the data pipeline especially when datasets and models are large and constantly evolving.
>
> - **Towards understanding the mechanism of instruction-tuning.** The fact that a base LM can be finetuned to follow instructions with extremely lightweight interventions that worth no more than 0.0019% - 0.0039% of the original parameter counts is surprising. Our finding can shed lights on uncovering the training dynamics of instruction-tuning.
>
> ### Clarifications on the prefix and suffix tokens.
>
> In the current draft, we experiment with a simple intervention strategy: intervening only on **the leading tokens** (“prefix” or the first $n$ tokens) and **the trailing tokens** (“suffix” or the last $n$ tokens) of the input prompt (i.e., there is no intervention being applied to output tokens). For classification tasks, we only intervene on the prefix tokens.
>
> The **intuition of this simple strategy** is that interventions on the prefix tokens change the “information read-out” of all following tokens working as anchors (since attention processes them differently now), and the interventions on the suffix tokens steer the generations.
>
> ### ReFT with DPO and the usage of the preference dataset Ultrafeedback.
>
> Indeed, DPO (or any arbitrary loss function) can be integrated with **a single model loaded into the memory** (taking the advantage of ReFT by only training the interventions). Additionally, we have integrated DPO trainers in our library which will be open-sourced.
>
> And yes, you are right - we are under-utilizing these preference dataset at this point. We only do SFT with the best rated responses without using the contrastive signals.

---

> > ### Comment · Reviewer_Pcei · 2024-08-13
> > **Reply to the rebuttal**
> >
> > I appreciate the clarifications the authors made around my questions. I like the one that mentioning quicker iteration for data selection on alignment pipeline using ReFT.
> >
> > I would keep my overall positive feedback on the paper and will keep my score unchanged.

---

### Author Rebuttal · Authors · 2024-08-04

We thank all reviewers for their useful comments. We remark on some of the shared questions here. All other questions are addressed in individual reviewer responses.

## Re: The significance of ReFT over LoRA and others.

Although in almost all other responses, we focus on comparing ReFT with other PEFTs by judging who achieves the SoTA performance, the most significant and surprising insight that we think ReFT brings to the table is **“minimally manipulating representations of LLMs can achieve strong finetuning performance”**.

ReFT also offers **a new generic finetuning paradigm** under which more variants could emerge as the community starts to explore the “steering” power of LLMs’ hidden representations. We showed some of the explorations in our Appendix F-H, yet there are much more to study both as research directions as well as practical applications: How to automate hyperparameter search for intervention locations? How to design a better intervention function? Is the intervention latent space interpretable? How to serve finetuned models with ReFT at scale? How to allow quick personalization for LLMs with ReFT? etc..


## Re: Remarks on hyperparameter selection with ReFT.

Multiple reviewers ask about the hyperparameter selection process with ReFT.

We want to clarify that **ReFT actually has a similar amount of hyperparameters to LoRA and other PEFTs**. ReFT only introduces two new hyperparameters, the intervening layers and the intervening positions, while *removing LoRA hyperparameters such as alpha and applied components (i.e., which component to apply LoRAs on) for LoRA*. We will clarify this further in the camera-ready. We also agree that it would be best if we could automate this process in the future. For now, we provide a practical guide of choosing hyperparameters for ReFT in Appendix D.2..

## Re: ReFT with other LLMs, such as Mistral and Phi-3.

Some reviewers raise the question regarding whether ReFT works with other model types.

To give a quick preview of ReFT with Mistral and Phi-3, we reran our quick adaptation experiments in Appendix G.2 with Mistral-7B-instruct-v2.0 and Phi-3-small-8k-instruct, where we finetune these two models to kindly refuse all user queries with 5 training examples with a rank=4 LoReFT on the last token at layer 15. **We include qualitative results in our attached rebuttal pdf**.

Additionally, we finetune various models from the Mistral and Phi-3 families as well as LLaMA-3-8B-Instruct on our math benchmarks with DiReFT. We use the same set of hyperparameters mentioned in the paper without additional hyperparameter searching.

| Name                     | % Params |  AQuA | GSM8K | MAWPS | SVAMP | Avg |
|--------------------------|----------|------:|------:|------:|------:|----------:|
| LLaMA-7B                 | 0.031%   | 21.3% | 24.1% | 74.5% | 42.7% |     40.6% |
| Mistral-7B-v1.0          | 0.031%   | 24.0% | 53.2% | 85.3% | 64.0% |     56.6% |
| LLaMA-3-8B-Instruct      | 0.031%   | 31.9% | 68.8% | 88.7% | 78.0% |     66.8% |
| Mistral-7B-Instruct-v0.2 | 0.031%   | 30.7% | 55.2% | 83.2% | 69.0% |     59.5% |
| Phi3-small-8k-Instruct   | 0.031%   | 37.0% | 79.8% | 92.0% | 84.1% |     **73.2%** |

These results definitely cannot provide the full picture without comparing against PEFTs. However, it is clear that ReFT (LoReFT and DiReFT) works for models from the Mistral and Phi-3 families. Bonus: Phi3 is obviously the best here as we expected since it’s actually tuned heavily on math and reasoning tasks!


## Re: Ways to improve ReFT on Math, and a showcase of ReFT with long-context summarization.

Multiple reviewers raise the question about what tasks ReFT is more suited for, and how to improve performance in benchmarks such as math reasoning.

We agree that LoReFT currently underperforms in math reasoning tasks compared with LoRA. One potential reason is that we only apply ReFT to the prompt tokens but not at generation tokens, which trades off the steering power for ReFT for lowered inference overhead. To verify this hypothesis, we ran additional experiments by **applying ReFT to selected prompt tokens as well as ALL decoding tokens** (LoReFT w/ decoding). Given our limited resources, we only ran it on our math reasoning benchmarks for a single seed with LLaMA-7B:

| Name                        | % Params |  AQuA | GSM8K | MAWPS | SVAMP | Avg |
|-----------------------------|----------|------:|------:|------:|------:|----------:|
| LoRA                        | 0.826%   | 18.9% | 37.5% | 79.0% | 52.1% |     46.9% |
| LoReFT                      | 0.031%   | 21.4% | 26.0% | 76.2% | 46.8% |     42.6% |
| LoReFT w/ decoding | 0.062%   | 20.1% | 31.2% | 80.7% | 54.8% |     46.7% |

This essentially closes the gap with LoRA. In addition, we note that this result is likely far from optimal: we ran this experiment without hyperparameter tuning, and we applied interventions on the attention output, not the residual stream. The additional parameters came solely from the decoding step interventions.

**Reviewer d959 brought up another related point: how applicable is ReFT for tasks requiring long-form context?** To address this with limited resources, we ran a validation experiment by applying ReFT to do long-context summarizations where we finetuned LLaMA-7B with only 10 examples sampled from the WikiSum [2] (input > 1000 tokens) and a single rank=4 LoReFT intervention (# of trainable params=32,772). We compared our model with an publicly available SFT summarization model. **We include one example in our attached rebuttal pdf**. ReFT can certainly adapt our base LM to do long-context summarization! In the next revision, we might consider adding a full-fledged experiment.

[1] Meng et. al., 2022, “Locating and Editing Factual Associations in GPT”

[2] Cohen et. al., 2021, “WikiSum: Coherent Summarization Dataset for Efficient Human-Evaluation”

---

### Decision · Program_Chairs · 2024-09-25

**Decision:**

Accept (spotlight)

**Comment:**

This paper proposes representation finetuning for parameter efficient fine-tuning.

Strengths:
1. The proposed representation fine-tuning is novel and efficient.
2. The paper demonstrates 15x to 65x improvement over LoRA. The experimental tasks are comprehensive. They also include three models.

Weakness:
1.  The cost of hyperparameter tuning is a concern.

The remaining concerns are already addressed well in author's response.

Overall, the paper proposes a novel method for parameter-efficient tuning and demonstrates its effectiveness across multiple tasks on three models. The improvement is impressive.